# Integrated Surveying, from Laser Scanning to UAV Systems, for Detailed Documentation of Architectural and Archeological Heritage

**Daniele Calisi *** , **Stefano Botta and Alessandro Cannata**

Department of Architecture, Università degli Studi Roma Tre, 00153 Rome, Italy; stefano.botta@uniroma3.it (S.B.); ale.cannata1@stud.uniroma3.it (A.C.)
* Correspondence: daniele.calisi@uniroma3.it

**Abstract:** Nowadays, the study and digitization of historical, architectural, and archaeological heritage are extremely important, covering the creation of digital twins—virtual replicas of real spaces and environments. Such reconstructions can be achieved using technologies with passive or active light sensors: laser scanners as light emitters, or photogrammetry through the creation of photographic images. As for the latter case, a distinction must be made between terrestrial and aerial shots, increasingly facilitated by the spread of UAV systems. Point clouds are aligned using georeferenced points measured with a total station. To create a faithful virtual model of the subjects, dense point clouds from a laser scanner are used to generate meshes, which are textured in high resolution from aerial and terrestrial photographs. All techniques can be integrated with each other, as demonstrated through the experiences of two case studies, each serving different purposes. The first is a detailed survey conducted for CAD representation of certain areas of Rocca Farnese in Capodimonte. The second is an instrumental survey for the creation of a realistic digital twin, aimed at providing an immersive VR experience of the archaeological area of Santa Croce in Gerusalemme in Rome.

**Keywords:** survey; photogrammetry; UAV; laser scanner; virtual reality; digital twin; restoration; archeology; architecture

## 1. Introduction

The development of survey techniques has ancient origins, evolving in response to the needs of its time. It began with the earliest land surveying practices to define agricultural areas, progressing through tools such as pretorian tables, which were used from the late 16th century for surveys even on a large scale by targeting lines through the instrument. This evolution continues to the present day, encompassing the latest digitalization of archaeological and architectural artifacts using both terrestrial and aerial instruments. This progression has occurred in constant dialogue with developments in other scientific fields, such as optics and geometry, resulting in continuous refinements and innovations. Notable studies include the development of stereoscopy from the 17th century and the construction of lasers in the 1950s [1].

To date, numerous methods and tools for digital surveying have been developed, each with varying costs and characteristics. Their particularity, compared with traditional techniques, lies in their ability to produce 3D models of captured subjects as outputs, characterized by a high fidelity to the original. Many advanced surveying tools fall into the category of contactless devices with the surveyed subject (Figure 1), a feature which makes them suitable for the preservation needs of areas such as archaeological areas. In particular, photogrammetry belongs to systems which acquire and process data from the passive reception of light radiation, in contrast to laser scanning devices like LiDAR, which actively emit radiation and then analyze the reflected echo data for surveying purposes [2].

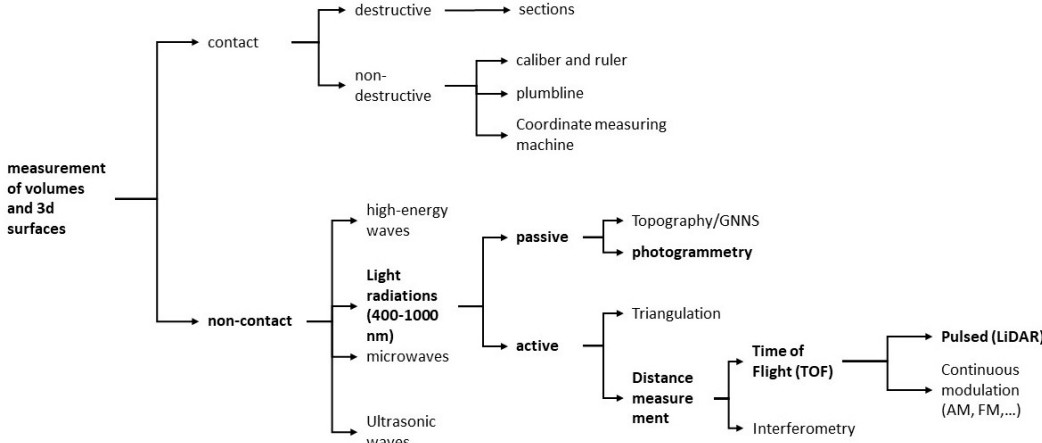

**Figure 1.** Taxonomy of measuring instruments in traditional and technological surveys with contact and non-contact techniques (image by the authors).

The ability to directly digitize architectural and artistic artifacts makes it possible to work with multi-scalar information, ranging from the smallest details to urban complexity. This has become an essential tool for archiving, preserving, and studying cultural heritage, as it facilitates the creation of databases of digital replicas capable of enhancing and assisting their analysis, dissemination, and management [3].

Among these techniques, photogrammetry refers to the principles of stereoscopy and the perceptual reconstruction of three-dimensional structures based on motion, supporting spatial detection and reconstruction. The photogrammetric surveying technique makes it possible to determine the shapes, sizes, and positions of objects in space through analysis, currently carried out by specific software, of at least two photographs of the subjects taken from two distinct points (known as a stereo pair). This process allows for the recognition of corresponding points, from which the scene can be reconstructed from different angles. The movement of the camera in relation to the object is crucial for deducing its three-dimensionality, which is why the technique is defined as structure from motion (SfM). For this reason, photogrammetric captures must occur in sequences of shots with uniform characteristics, depicting the object with a certain overlap percentage (at least 50%, though it might be variable) to allow for the recognition of homologous points within the set of images [4,5].

The idea of using photography as a measuring tool began to take shape around 1850. It started with Arago, the Director of the Paris Observatory, who requested specific resources from the French Chamber of Deputies to develop photographic techniques for producing increasingly accurate topographic maps [6,7]. In 1849, Colonel Aime Laussedat demonstrated the usefulness of photography for creating topographic maps. The first shots for this purpose were taken from the tops of mountains or the roofs of particularly tall buildings. Later, equipment was installed on early hot air balloons (Figure 2). The technique of obtaining measurements from a photograph was initially called "metrophotography" and was later referred to as photogrammetry, a term coined by the German engineer Meydenbaur in 1870 [8]. Having already applied intersection calculations to perspective views using the "camera lucida" (an optical device used for accurate drawing of viewed subjects), Lussedat later applied it to photographs of urban and mountain landscapes. His demonstrations in Paris and Buc, near Versailles, in 1861, convinced the French army to establish a specialized brigade for the development of photogrammetry [9].

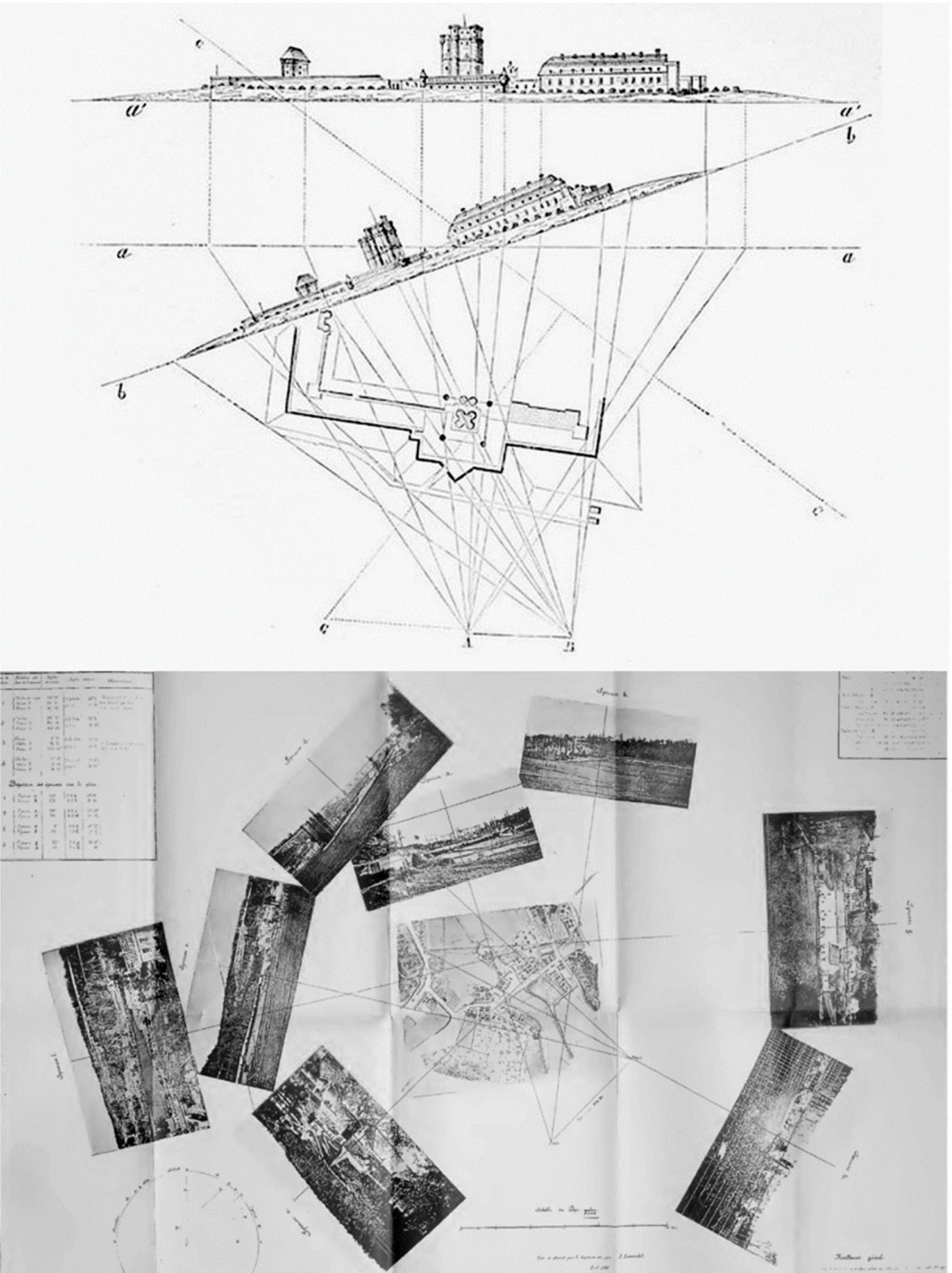

**Figure 2.** From the top: Laussedat, reconstruction of the facades of the Vincennes Fort in Paris carried out in 1850; Laussedat, photographic survey of Buc near Versailles carried out in May 1861. The eight constituent photographs of the survey, placed near the recording stations, with the developing map in the center [8].

Exploited by numerous scientific disciplines, photogrammetric techniques can be divided into two main categories: aerial and terrestrial photogrammetry. The distinction lies in the position of the camera and the distance of the sensor from the surveyed object, although the quality of graphical results obtained from the two photogrammetric modes can sometimes be similar.

Terrestrial photogrammetry employs primarily static devices through which sequential shots are taken using different capturing techniques, mostly keeping a close distance from the subject. There are a range of devices with different uses and levels of precision. This

includes metric cameras, characterized by specialized lenses to control distortions and internal orientation systems, as well as amateur cameras, which can still achieve good resolutions.

Aerial photogrammetry, on the other hand, exploits devices such as drones to capture areas hardly accessible from the ground, and also optimizes results for subjects typically reconstructed by terrestrial survey. The advantages of unmanned aerial vehicles (UAVs) also lie in reducing acquisition time and costs, minimizing the number of shots required, and lowering risks for operators, who can control the flight remotely from safe areas, even in situations of inaccessibility or other hazards [10]. UAVs can be equipped with various types of sensors, making them versatile surveying tools across multiple sectors. The propulsion systems of the flying device are equally crucial in executing an aerial photogrammetric campaign. Fixed-wing drones, for instance, are more suitable for large open areas, requiring more takeoff and landing space. They cannot move at low speeds or get too close to the subject being surveyed. Helicopters and multirotor drones are commonly employed in this field, as they are easy to maneuver in all directions and can stably hover in place. There are also less common hybrid systems which allow for conversion between vertical and horizontal propulsion, making them suitable for various situations [11].

It is possible to determine the achievable level of detail with a device in advance, based on its technical specifications: the pixel size of the sensor and the focal length used for capturing make it possible to calculate the minimum visible size of an object in the image, depending on the distance from the object. This is known as ground sample distance (GSD) [5]. Using a simple proportion (Figure 3), it is easy to determine the maximum acceptable distance between the device and the surveyed scene to achieve a specific representation scale. In these cases, photographic datasets are composed by planning the flights over the analyzed areas, which is extensively useful for surveying large surfaces, where it is important to plan the percentage of overlap between photographs based on the altitude and the spacing between parallel flight paths executed by the drone.

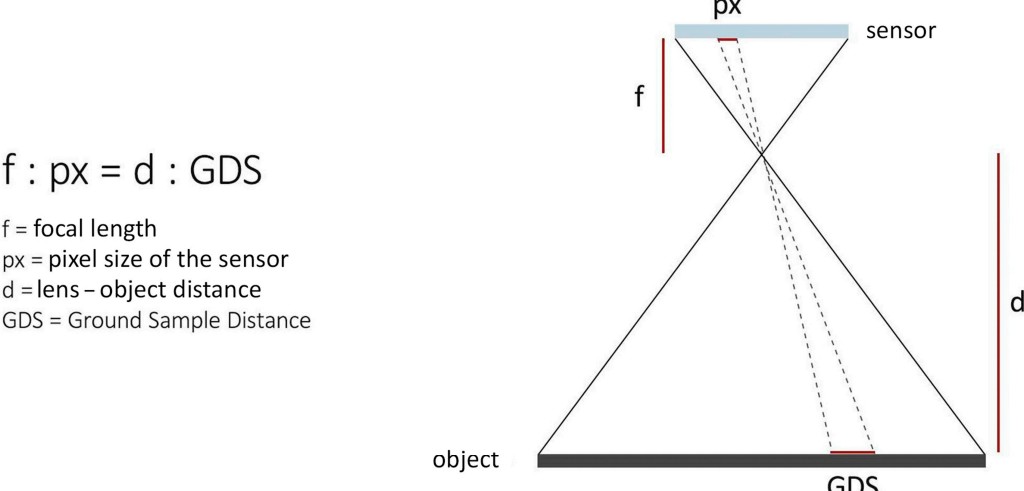

$$f : px = d : GDS$$

f = focal length
px = pixel size of the sensor
d = lens – object distance
GDS = Ground Sample Distance

**Figure 3.** Definition of GSD (ground sample distance) in relation to the characteristics of the camera and its distance from the detected object. Based on the GSD it is possible to understand up to which scale of representation it is possible to represent the architectural object so that the perspective error of the restitution is acceptable (image by the authors).

Both techniques employ control points, which are positioned on the ground. In the case of aerial photogrammetry, these points are spatially identified through geographic coordinates and are usually defined using GPS or GNSS equipment, which provide centimeter-level precision in spatial coordinate determination. In terrestrial photogrammetry, these points are placed directly on the object being surveyed to facilitate measurements between different targets. These control points are also crucial as references in the subsequent alignment processes among different types of surveys, allowing the algorithms of the pho-

tomodeling software to recognize and align datasets with each other. This way, it is possible to combine aerial and terrestrial photogrammetric techniques, achieving not only complete reconstructions of artifacts but also maintaining different scales; this is particularly useful for structuring detailed models of extensive areas [12].

Digital acquisition can lead to various two-dimensional and three-dimensional graphical elaborations (georeferenced or not), including point clouds, meshes, and orthophotos [13]. For instance, digital terrain models (DTMs) and digital surface models (DSMs) are products of this technology: the former focuses on the land surface with information related to the topography of the ground plane, excluding all objects in the area; the latter represents the surface as captured, including vegetation, urban structures, and all anthropic interventions [14].

The advantages of both aerial and terrestrial photogrammetry lie in the speed and efficiency of data acquisition and processing. Typically, information supporting territorial and urban planning is updated every 10–15 years. Using photogrammetry, cartographic databases for local-scale projects can be updated in a shorter time frame [15].

New integrated survey technologies have allowed the analysis and study of archaeological, historical, and architectural heritage much more accurately. Public administrations and institutions have gradually begun a process of virtual digitization of assets under their responsibility. They understand the importance of having access to easily consultable digital twins for the management, visualization, cataloging, and programmatic planning of interventions for the conservation of the heritage itself. The case studies are numerous, and, at an international level, each case obviously has its specificity and the approach to each site is always different based on the geo-morphological characteristics of the artefact to be surveyed. Furthermore, it is also necessary to diversify the case studies based on the ultimate purpose of the survey itself: from the mere creation of a digital copy, to the mapping of materials and degradations for the purpose of conservative restoration, to visualization in a virtual environment through XR, up to immersivity in the site and in its context [16–20].

The presented research proposes an analysis of integrated digital survey processes and methodologies involving both aerial and terrestrial photogrammetric techniques in the detailed reconstruction of architectural and archaeological artifacts, in conjunction with other surveying tools like laser scanners. The aim is to obtain faithful replicas of reality, which can serve as the basis for further bidimensional and three-dimensional studies of the surveyed subjects. The purpose is not only to evaluate geometric and material qualities, which are essential in fields like restoration, but also to address data management issues arising from different types of instrumental surveys. Despite their different goals, the applied methodology is retraced through the two presented case studies. This encompasses the stages of survey and data capture, as well as the development and alignment of point clouds, leading to the production of the final models. The first case study focuses on Rocca Farnese in Capodimonte (VT, Italy), a 13th-century fortress on Lake Bolsena. The integrated survey aims to provide 2D–3D documentation for the study and restoration of the artifact. The second case study concerns the archaeological area of Santa Croce in Gerusalemme in Rome (RM, Italy). Aside from its analytical purposes, the project aims to develop an interactive and immersive virtual reality experience.

## 2. Materials and Methods

The first case study is the result of a collaboration with architect Anelinda Di Muzio on behalf of the Soprintendenza Archeologia Belle Arti e Paesaggio per la Provincia di Viterbo e per l'Etruria Meridionale. The subject is Rocca Farnese in Capodimonte, located in the province of Viterbo, Italy (Figure 4). Situated in the center of the Tuscia, in an area which preserves the memories of the Etruscans and the Renaissance residences of Caprarola, Vasanello, Vignanello, Bomarzo, Villa Lante, and Capodimonte, Rocca Farnese plays a significant role as a testament to the region's rich cultural heritage. Originally constructed as a fortress in the 13th century by the ancient nobility of Bisenzio, it was later transformed into a palace in the 16th century, for which the project was assigned to Antonio da Sangallo

il Giovane, who emphasized its quadrangular plan. The building is situated on top of the historic center, on a volcanic cliff overlooking Lake Bolsena. In the past, the Rocca was the residence of the noble Farnese family and hosted some of the most illustrious popes in history. I also became a representative residency of Duchy of Castro, until its decline in 1649. Nowadays, Rocca Farnese is a private property of multiple owners but it's still partially visitable.

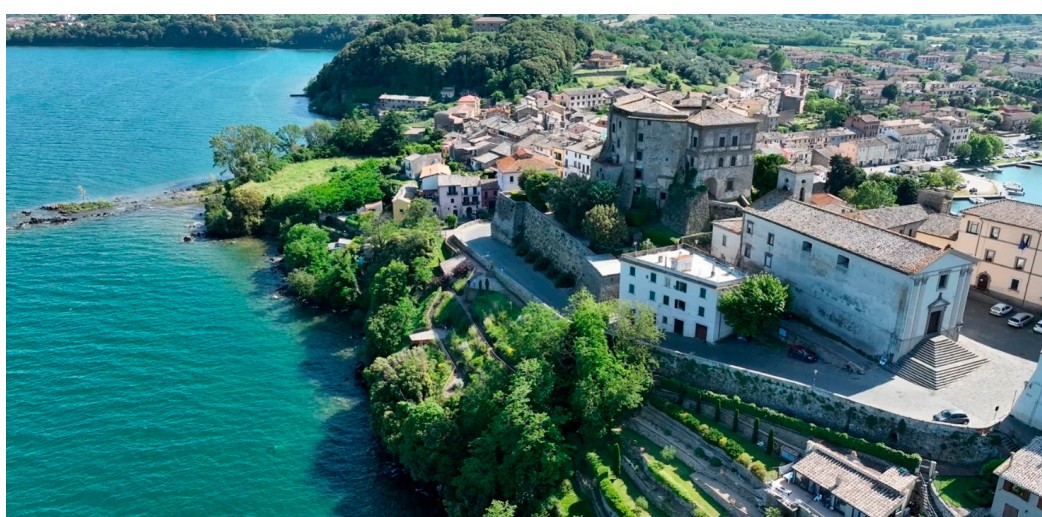

**Figure 4.** Aerial view of Rocca Farnese in Capodimonte on the Bolsena Lake. The small village is dominated by the Rocca which greatly dominates the entire town. The defensive walls containing the palace gardens are also significantly important (image by the authors).

The restoration of the Rocca Farnese in Capodimonte and all related activities, including the survey of the monument, are funded by the European Union, PNRR-CAPUT MUNDI, NEXT GENERATION EU IN ROME, Mission 1 Component C3, Ministry of Tourism subject holder of the funding, Soprintendenza Archeologia Belle Arti e Paesaggio per la Provincia di Viterbo e per l'Etruria Meridionale subject implementing the funding.

The survey of the Rocca was carried out using two different methodologies: one conducted using a laser scanner, the other involving aerial photogrammetry with various drones, different in sizes and characteristics.

During the initial phase of the campaign, which was crucial for the execution of the integrated survey, various types of targets were positioned. These targets served as reference points for the topographic survey, aerial photogrammetry with UAVs, and the laser survey. These targets played an essential role in ensuring the accuracy and consistency of geospatial data collected during the entire surveying process. The activities related to the topographic and drone surveys required distinct targets, considering the different techniques used and the specific goals of each operation. For the topographic survey, small metal plates were strategically placed in the surrounding environment. These plates were attached to structural elements, such as walls, pillars, and poles. This allowed for the establishment of a reference coordinate system and ensured a precise georeferencing of collected images and data. For the drone survey, a total of 124 targets were used, known as photogrammetric control points. To maximize operational efficiency, predefined targets were directly exported from Agisoft Metashape Professional 1.7.2 and then physically placed in the site. This choice was made to facilitate recognition by the same software, used later in the process to generate the point cloud. These targets consisted of reference images with specific geometric patterns. The shape and distribution of these patterns allowed the photogrammetric software to easily recognize them in the photos acquired by the drone. Accuracy in positioning of the reference points significantly influenced the overall precision of the survey and the subsequent cartographic processing. Furthermore, it was crucial to create detailed documentation regarding the position of each target, including

coordinates and information about the surrounding environment. This facilitated the post-processing phase of the data. The topographic survey was then conducted to georeference characteristic points and determine their spatial coordinates (X, Y, Z) within a national reference system by a total station connected to a GNSS. Subsequently, this information was applied to the aerial photogrammetric survey to identify and associate common points between the two datasets.

During this phase, a survey campaign was also conducted using the Z+F IMAGER 5010 laser scanner (Figure 5) for the spaces in front of the main facade, the bridge and the area beneath it, the courtyard, and all interior rooms on the ground floor, including access points to adjacent levels.

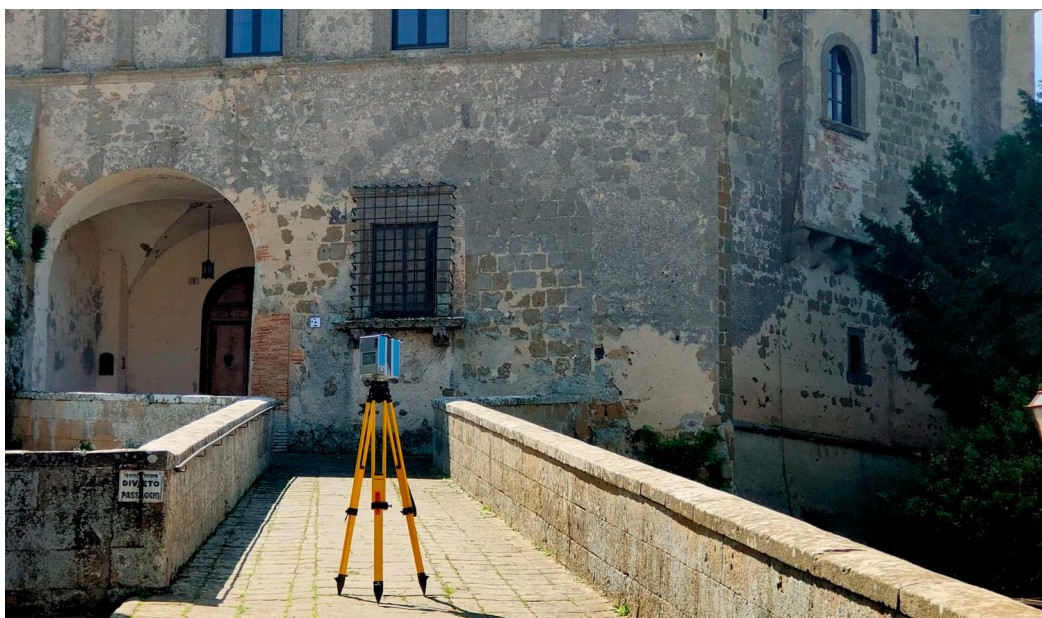

**Figure 5.** Photo taken during the survey campaign, showing the laser scanner employed and one of the targets located on the wall next to the entrance arch. Different quality levels were used during the scanning process, mainly based on the distance between the artifact and the device. In particular, external scans were set to a higher-resolution quality to obtain greater accuracy of the data (image by the authors).

The laser scanner survey project was previously planned on a historical floor plan provided by the client, aiming at minimizing blind areas and adjusting the number of stations to ensure visibility between different positions for an appropriate alignment of point clouds, with minimal and acceptable margins of error. A total of 103 scans were required, mostly at medium resolution due to the relatively small size of the interior spaces (Figure 6). The presence of furniture and a significant number of windows resulted in numerous blind spots and noise caused by laser refraction and reflection on surfaces such as glass and mirrors. However, this did not affect processing, as the resulting point cloud was only meant for 2D redesign of the interiors (minimizing blind areas due to occlusions) and for understanding wall thicknesses; there was no need to fully recreate the furnished interior in the final output.

Photography played a crucial role in the architectural survey conducted with the assistance of drones. Exploiting modern technologies and the flight capabilities of these devices, it was possible to capture unique and distant angles, hardly accessible otherwise. The drones used during the survey activities included the DJI Mavic 3 Cine with an L2D-20c camera, the DJI Inspire 2 with an HG310 camera, and the DJI Mini 3 Pro with an FC3582 camera. The use of these tools allowed for the integration of information acquired during terrestrial survey campaigns, capturing the complexity of this particularly wide architecture, with resolutions of up to 5280 × 3956 pixels. Simultaneously, a photographic campaign

was conducted for the interior rooms of the ground floor, the courtyard, the outdoor spaces above and beneath the bridge, and in front of the main facade. The dataset was acquired with a Canon EOS 6D camera with a Canon EF 24–105 mm f/4 L IS USM lens (mostly set at a focal length of 24 mm, with variable zoom for detailed shots of decorations). A total of 1270 RAW photos were taken and cataloged according to 27 areas to be easily checked during the CAD redrawing phase. This resulted in a wide archive of references, including an additional 160 photos taken by smartphone (Huawei 30 Pro, Leica Quad Camera, with a Super Spectrum CMOS main sensor and stabilized 27 mm f/1.6 OIS lens), as well as around 20 videos in MP4 and AVI by drone.

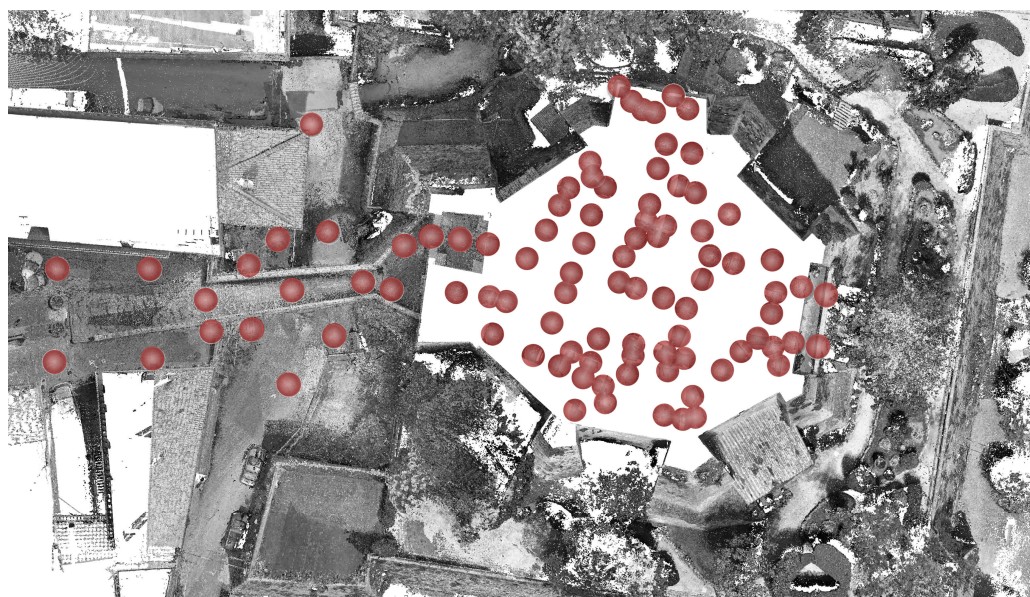

**Figure 6.** In red, the positions of the laser scans according to the subsequently processed point cloud in Autodesk ReCap. The laser scanner survey included the ground floor, the main facade on the north, the access bridge, and the surrounding context (image by the authors).

The availability of a wide set of photographs allowed for a comprehensive view and a deep control of Rocca Farnese, including details which were difficult to reach. To ensure optimal results in photogrammetry, a meticulous calibration of the cameras was necessary, considering lighting conditions and reflections in the surrounding environment (Figure 7). Additionally, ground-based reference points and targets were used for image georeferencing.

The entire drone survey operation involved a total of 8567 photographs, of which only 3602 were later included on Agisoft Metashape Professional 1.7.2. Among these 3602 images, 1371 shots were taken by the L2D-20c camera, characterized by a focal length of 12.29 mm, a resolution of 5280 × 3956 pixels, and a pixel size of 3.36 × 3.36 µm. An additional 97 photographs were acquired using the HG310 camera, featuring a focal length of 3.61 mm, a resolution of 4000 × 2250 pixels, and a pixel size of 1.7 × 1.7 µm. Lastly, 2134 images were captured by the FC3582 camera, with a focal length of 6.72 mm, a resolution of 4032 × 3024 pixels, and a pixel size of 2.4 × 2.4 µm. During the photograph alignment phase, some images were not recognized by the software, resulting in a final count of 3204 successfully aligned cameras.

To begin with the alignment process, the initial step involved the manual identification and recognition of targets from the topographic survey in all the drone photographs. By assigning spatial coordinates (X, Y, Z) to each of these targets, a rigid grid of fixed topographic points was effectively established, which was later considered by the software during the construction of the model. Subsequently, certain technical parameters were set, including the accuracy level being set to "High" to ensure the best result available in align-

ing photographs, fully exploiting the capabilities of the software. The accuracy parameter is directly proportional to processing time, but inversely proportional to the desired margin of error during this phase. Processing time is influenced by the PC specifications: in this case, a laptop with a 64-bit Windows operating system, equipped with 127.77 GB RAM, a 13th Gen Intel(R) Core(TM) i9-13900K CPU, and an NVIDIA GeForce RTX 4090 GPU.

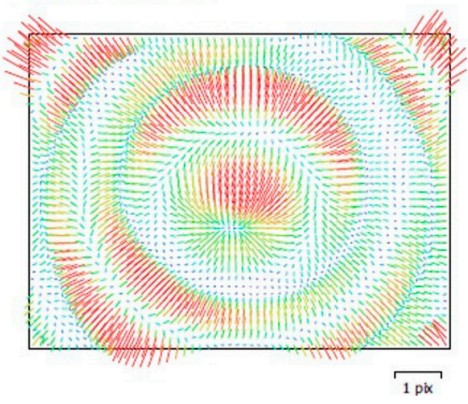

## Camera Calibration

**L2D-20c (12.29mm)**
1371 images

| Type | Resolution | Focal Length | Pixel Size |
|------|-----------|--------------|------------|
| **Frame** | **5280 x 3956** | **12.29 mm** | **3.36 x 3.36 µm** |

| | Value | Error | F | Cx | Cy | B1 | B2 | K1 | K2 | K3 | K4 | P1 | P2 |
|---|---|---|---|---|---|---|---|---|---|---|---|---|---|
| **F** | 3728.31 | 0.011 | 1.00 | 0.03 | -0.20 | -0.24 | -0.00 | -0.48 | 0.46 | -0.42 | 0.38 | 0.02 | -0.22 |
| **Cx** | -6.868 | 0.022 | | 1.00 | 0.01 | -0.04 | 0.04 | 0.00 | -0.00 | -0.00 | 0.00 | 0.95 | 0.01 |
| **Cy** | 20.86 | 0.02 | | | 1.00 | -0.02 | -0.04 | -0.01 | 0.00 | 0.00 | -0.00 | 0.02 | 0.93 |
| **B1** | 0.417082 | 0.0065 | | | | 1.00 | -0.01 | -0.00 | -0.00 | 0.00 | -0.00 | -0.03 | 0.09 |
| **B2** | 0.312897 | 0.0062 | | | | | 1.00 | -0.00 | 0.00 | -0.00 | 0.00 | -0.05 | -0.01 |
| **K1** | 0.0366065 | 2.9e-05 | | | | | | 1.00 | -0.97 | 0.93 | -0.88 | 0.00 | -0.01 |
| **K2** | -0.187408 | 0.00016 | | | | | | | 1.00 | -0.99 | 0.96 | -0.00 | -0.00 |
| **K3** | 0.405141 | 0.00036 | | | | | | | | 1.00 | -0.99 | 0.00 | 0.00 |
| **K4** | -0.264526 | 0.00027 | | | | | | | | | 1.00 | -0.00 | -0.00 |
| **P1** | -0.000817715 | 1.7e-06 | | | | | | | | | | 1.00 | 0.02 |
| **P2** | 0.000725016 | 1.5e-06 | | | | | | | | | | | 1.00 |

**Figure 7.** Calibration of the drone camera. In the table below, the calibration coefficients and correlation matrix (image by the authors).

Regarding the initial phase of the alignment process, called matching, it took 16 min and 47 s, with a memory usage of 3.42 GB. The second phase of the process, alignment, required a longer time, 1 h and 22 min, with a memory usage of 4.41 GB. The outcome of the alignment phase led the software to recognize the photographs in space and align them to create an initial sparse point cloud, consisting of 1,722,938 points (Figure 8).

The subsequent phase focused on generating the dense point cloud (Figure 9), with careful attention to setting specific parameters, such as configuring the quality to "High" and setting the filtering mode to "Aggressive." The process took a total of 6 h and 58 min, with a memory usage of 71.95 GB, and resulted in a file size of 7.64 GB. Initially, the point cloud consisted of 537,192,463 points, but it was later subjected to a cleaning and optimizing process to remove points which were irrelevant to the context of the surveyed area, as well as dense vegetation and shrubs which covered significant portions of the Rocca. The cleaning operation was carried out on each laser scan, in some cases also using the cloud clipping tool to focus on superfluous or incorrect parts (due to reflections or refractions) to avoid deleting useful parts of the survey. This operation aimed at avoiding issues during

two-dimensional reconstruction of the building's facades. As a result, the model was reduced to 181,914,544 points, significantly lowering the size of the project file and making it more manageable.

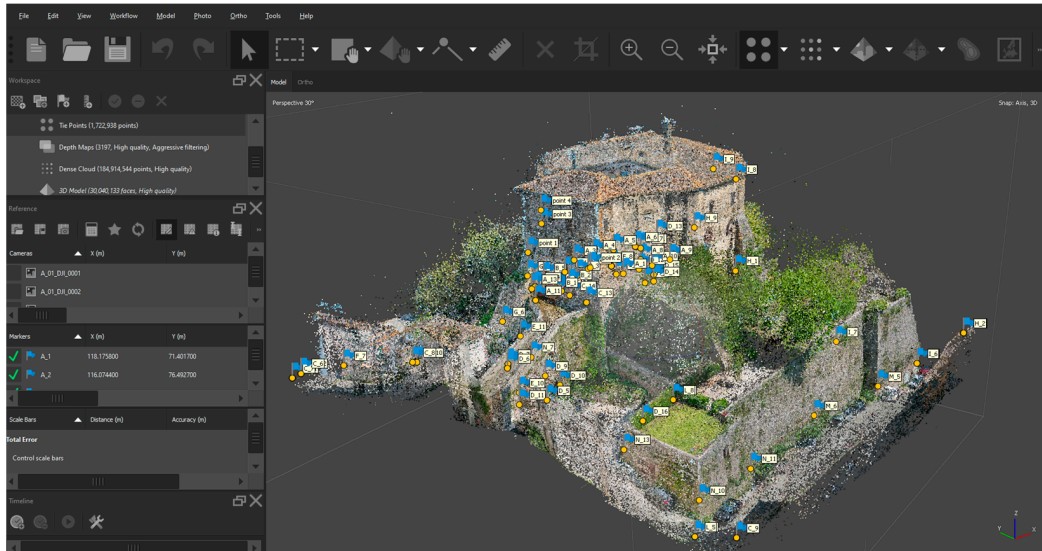

**Figure 8.** Sparse point cloud built in Agisoft Metashape. The creation of the sparse point cloud is the first step of the reconstruction, resulting from the alignment of the photographs in space. The alignment process was implemented by recognizing the targets whose real coordinates were previously taken by the total station connected to a GNSS (image by the authors).

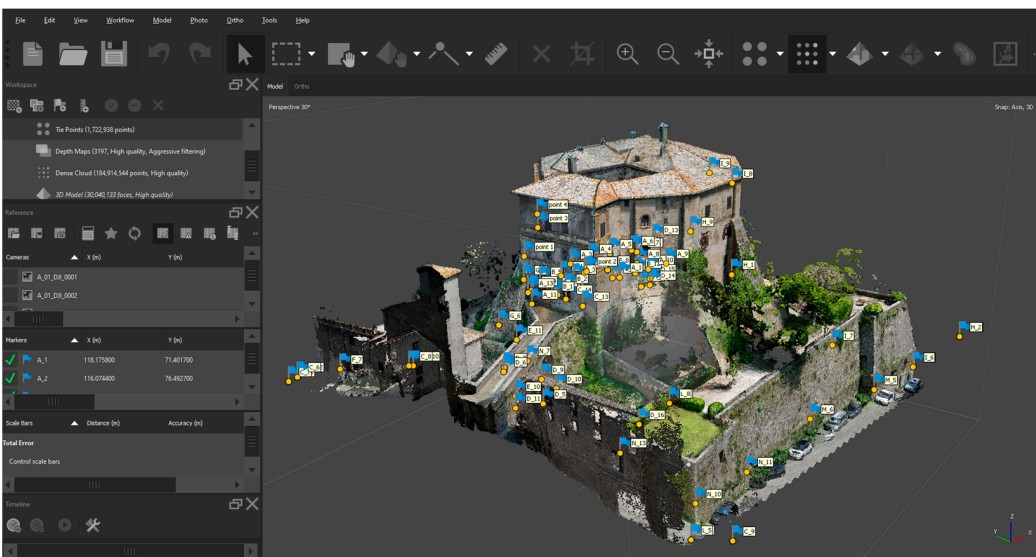

**Figure 9.** Dense point cloud built in Agisoft Metashape. The rigid grid defined by the 124 targets (with real coordinates taken from the total station connected to a GNSS) allowed for the alignment of all the frames of the virtual reconstruction with a minimal margin of error (image by the authors).

A 3D mesh reconstruction was then generated by starting the "Build Mesh" process. This operation led to the creation of a model consisting of 39,526,220 faces and 19,806,251 vertices, ready for the texturing process. Specific parameters were set during this phase, configuring the mapping mode to "Generic" and setting the blending mode to "Mosaic". Defining a texture size of 8192 × 8192 pixels was decisive to achieve an optimal level of detail in the mapping. The options for "Enable hole filling" and "Enable ghosting filter" were both enabled to ensure high-quality results in the process. The initial phase of the texturing process, known as UV mapping, took a total of 24 min and 41 s,

with a memory usage of 5.07 GB. The subsequent blending phase took 42 min and 37 s, utilizing 11.13 GB of memory. At the end of the texturing process, the file size increased by 1.90 GB in addition to the previous memory used in the earlier stages. During these processes, a second subgroup of points, called a "chunk", was created in parallel with the one used for processing, in order to preserve an intact copy of the dense point cloud, without any point removal. This operation proved useful during testing in the meshing (or texturing) process, as it made it possible to identify the areas of file optimization based on the final weight of the product undergoing texturing. During a later phase of the process, this mesh model was used, together with the complete point cloud (made by joining UAV and laser scanner models), to elaborate and export the orthophotos needed for the 2D reconstruction. In fact, simultaneously with the processing of the drone-based aerial photogrammetry, the scans obtained by laser scanning were imported and aligned using the Autodesk ReCap 2023 v.22.0 algorithm, obtaining the point cloud of the interior areas of the ground floor and the exterior areas close to the main fronts, above and below the access bridge. This point cloud consisted of a total of around 638,600,000 points (Figure 10).

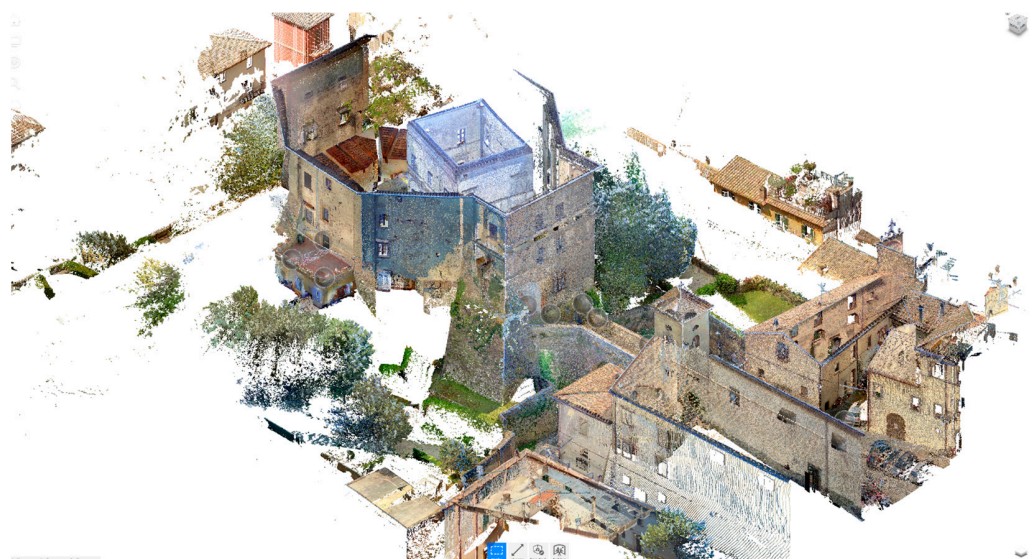

**Figure 10.** Point cloud processed in Autodesk ReCap from the laser scanner survey. The alignment of the scans starts from a master station which controls the entire project, to which the subsequent scans begin their alignment. The process is a progression: the second aligns with the first, the third with the second, and so on. Sometimes the source and destination scans need to be set manually, especially if the survey does not follow a chronological order, as often happens. The overlap is done automatically, or by indicating three pairs of points on two different scans (image by the authors).

The dense point cloud generated by the drone was simultaneously exported in E57 format to proceed with the alignment to the point cloud generated by the laser scanner. This model was then imported into Autodesk ReCap 2023 software within the project, where the laser scanner point cloud was already processed. Both point clouds were in real scale (1:1), as the drone-generated point cloud had already integrated the data obtained from the topographic survey (associated to the national reference system), while the laser scanner point cloud was already in real scale through its own reference system. However, the point clouds were in different positions of the virtual environment. Therefore, it was necessary to move and align the dense point cloud generated by the drone to the point cloud from the laser scanner; this was made by selecting the same set of points in both models, choosing three targets positioned within the courtyard. This procedure resulted in the loss of topographic data and the national reference system. However, such data were not essential for the purposes of the research, which included the two-dimensional representations of plans, elevations, and sections. After this operation, a complete point cloud was obtained, in which the information

acquired from the laser scans was integrated with the information from the drone survey. Before proceeding further, a verification was carried out to ensure that the divergence between the two point clouds was only 2 mm. Once the accuracy of the result was confirmed, the new complete point cloud was saved in RCP format.

To proceed with the two-dimensional reconstruction phase, the complete point cloud was imported into AutoCAD 2022 as an external reference and centered into the standard origin of the workspace. The next step was to define a section plane for the ground floor and for each elevation to be surveyed and redrawn. As for the ground floor, the reference z = 0.000 elevation of the workspace was taken to create a "slice" type section plane (to control not only the position of the cut but also its thickness, managing both sections and projections). This plane cut through the point cloud at a height of +1.25 m relative to the chosen building reference elevation (0.00), which was a point on the floor of the internal courtyard portico (nearly flat); this data was crucial for dimensioning the drawings, both plans and elevations. Starting from the section obtained from this plane, all vertical section planes ("slice" type) were positioned and sized (Figure 11), with attention to ensure their parallelism with their respective facades. This resulted in a floor plan and 16 elevations: three main external elevations corresponding to the entrance facades, four internal elevations corresponding to the courtyard, and two elevations of the access bridge, for which a 2D redrawing at a 1:50 scale was planned along with material representation by overlaying the orthophotos from the mesh model; seven elevations of the remaining external fronts, for which only the material representation through orthophotos at 1:50 was planned; and a ground floor plan at a 1:100 scale (including the entire garden and access bridge) and a 1:50 scale (focused on the Rocca), for which only the two-dimensional redrawing was planned. Since the work was done in 2D on a sectioned point cloud within a 3D environment, a specific user coordinate system (UCS) was defined for each section plane, placing the XY plane on it and the pointing Z in relation to the plane's reading direction. This approach not only allowed the elevation to be displayed correctly on the orthogonal projection coinciding with the section plane but also allowed projection on the XY plane of any drawn element (with Z elevation = 0 related to the chosen UCS) by activating the "replace Z value with current elevation" setting, thus facilitating and optimizing the reconstruction work. This CAD project was then used as a template for each drawing.

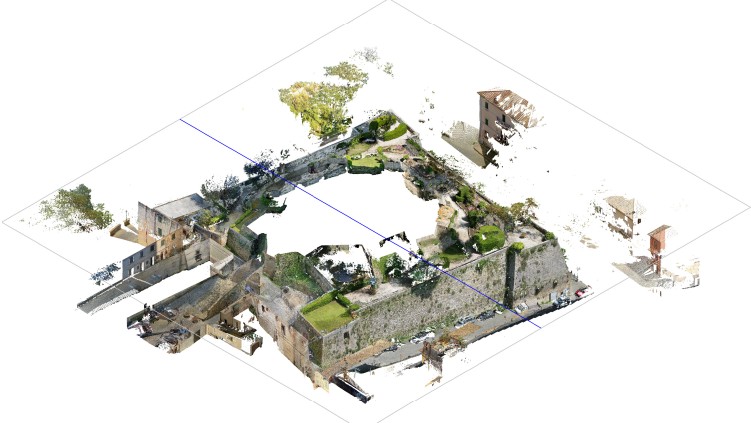

**Figure 11.** The point cloud produced by laser scanning was subsequently aligned with the point cloud deriving from photo-modeling in Metashape and exported in E57. The complete point cloud, here imported into AutoCAD 2022, is cut by the section plane created for the floor plan. Here, the section plane of one of the elevations is underlined in blue (image by the authors).

As previously mentioned, for 2D redrawing of the facades, an overlay with high-resolution orthophotos was planned to accurately represent material variations on the surveyed surfaces. For this reason, the meshes obtained earlier were not further cleaned or optimized, but directly exploited to create the images using the "Build Orthomosaic" tool

in Agisoft Metashape Professional 1.7.2. These orthophotos were developed to be aligned in reference to the section planes previously defined in the CAD environment on the point cloud, so they could later be placed in coincidence with them. This made it possible to work simultaneously on the sectioned point cloud and its corresponding raster image obtained from the mesh, providing a more detailed view of facade material qualities during the 2D redrawing phase; it also enhanced understanding of the structure and improved control over the three-dimensional aspects of the drawing. To obtain each facade, three markers were placed on the mesh to be recognized and used to set the horizontal and vertical axes of the projection plane. Using the mesh as the reference surface and enabling automatic hole filling, all orthophotos were generated (Figure 12), including the facades that did not require redrawing, with a pixel size of approximately 0.027 px/m (considering 300 DPI resolution images to be scaled at 1:50). These were exported in PNG format to maintain background transparency, with dimensions ranging from 3288 × 4096 px to 17391 × 10239 px.

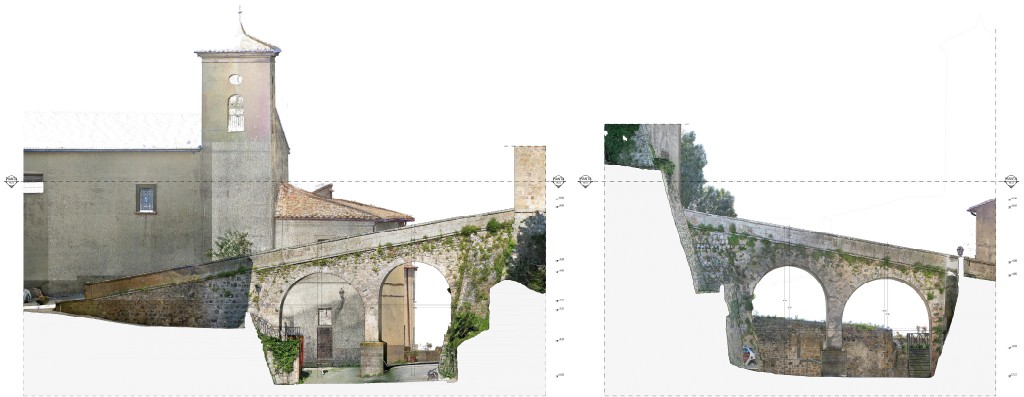

**Figure 12.** Elaborations showing the two sides of the bridge at the entrance. Here, the orthophotos generated were also integrated with parts from the point cloud where the mesh version of the model was missing (the church in the background of the image on the left, for instance). Then, the bridge was re-drawn in AutoCAD, considering the orthophotos, to convey a full and detailed representation of the manufact and its material characteristics (images by the authors).

Some of the orthophotos processed in Agisoft Metashape had issues related to the quality of the mesh; in particular, the edges of the facades facing the courtyard of the Rocca were unclear and difficult to read due to an inability to get the drone close to the internal corners to avoid the risk of collision. Therefore, it was decided to attempt a reconstruction of the interior orthophotos using another SfM program, specifically, Reality Capture Beta 1.0, to evaluate other results using a different algorithm. Following the same workflow, textured mesh models of the courtyard facades were processed and orthophotos were generated.

Each orthophoto was then imported as a raster reference into the respective file, placed on the correct section plane, carefully scaled, and aligned to the orthogonal view from the point cloud. For each redrawn facade, two versions of the same orthophoto were elaborated: the first at 100% opacity, to be overlaid by CAD drawing; the second with a white background under the redrawn areas, leaving the remaining context at 60% opacity to highlight the two-dimensional representation (Figure 13).

The final phase of the process was the 2D reconstruction of the surveyed floor plans and elevations (Figures 14 and 15), working in detail by integrating information derived from orthophotos, point clouds, and high-resolution photographs obtained from aerial and terrestrial campaigns. These were essential to understand details such as decorations, which were redrawn geometrically where possible, or in their current state due to the widespread degradation of certain architectural elements.

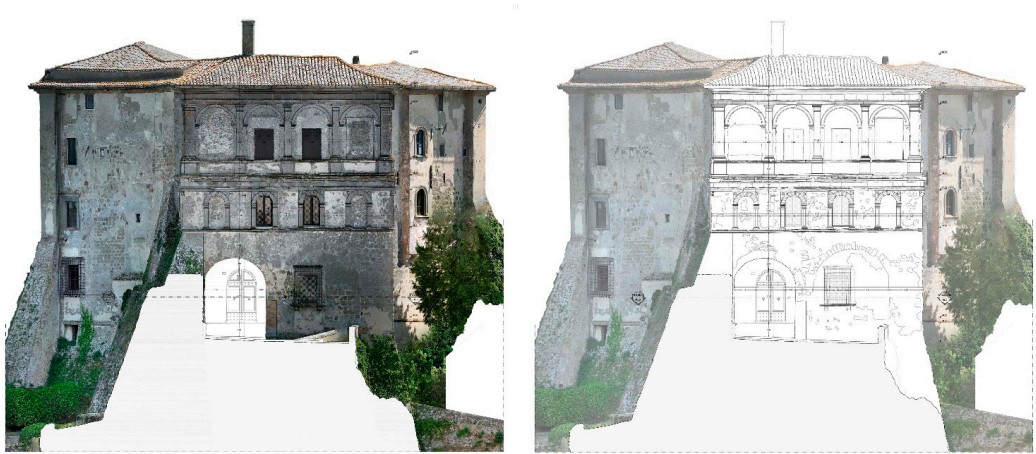

**Figure 13.** The two versions of the facade: on the left, the 2D representation overlaying the orthophoto at 100% opacity; on the right, the redrawn facade with a white background and the orthophoto at 60% opacity that completes the rest of the facade. The double representation was necessary to appreciate the quality of the details of the 2D drawings, which were the basis for an initial mapping of materials for the subsequent restoration project requested by the client, which analyzes degradations and interventions. At the same time, the full-opacity representation also makes it possible to appreciate the detail of the orthophoto in its material details (images by the authors).

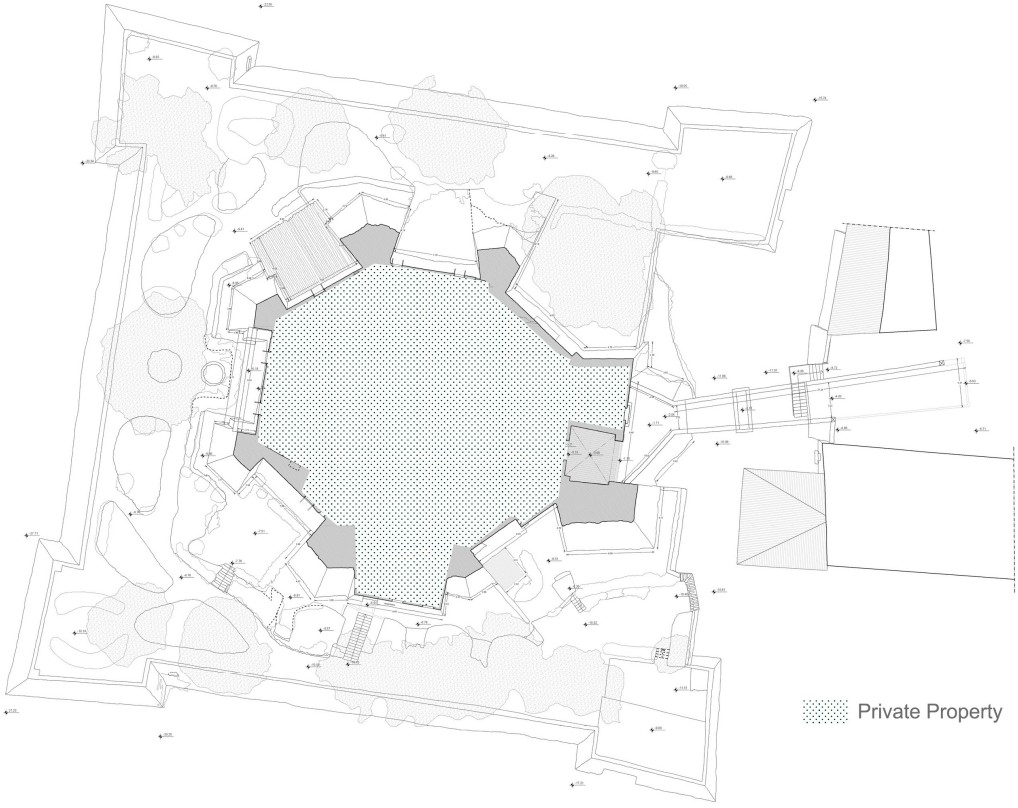

**Figure 14.** The 1:100 scale plan of the ground floor of Rocca Farnese, elaborated in AutoCAD 2022. The restitution process is complex, and it is also the result of a cross-check between the point cloud and the detailed photographic documentation, which allows for a better understanding of the various moldings and their generating geometries (images by the authors).

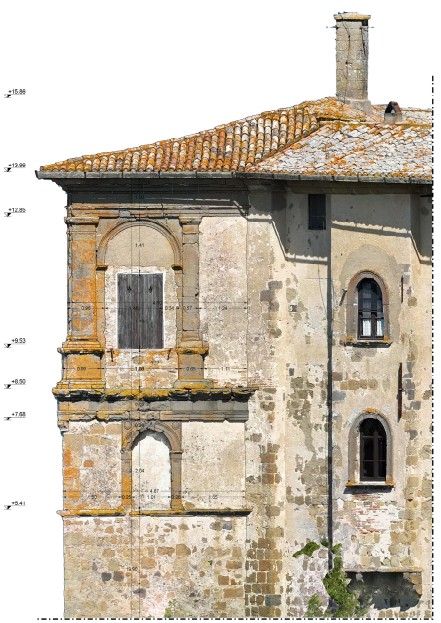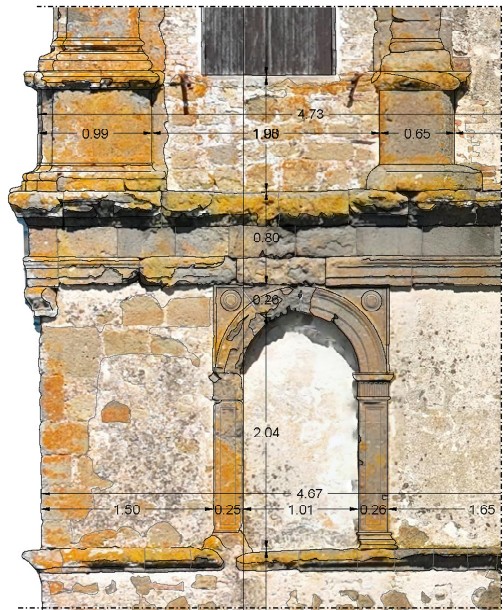

**Figure 15.** Parts of the elevation on the side of the main facade, redrawn from the integrated digital survey, overlaid on the orthophoto. On the right, a detailed focus of the same elaboration (images by the authors).

The second case study involves an integrated digital survey with different purposes, carried out at the archaeological site of Santa Croce in Gerusalemme in Rome (Figure 16). Located in the Rione Esquilino, the studied area is enclosed to the north by the Aurelian Walls and the Claudian Aqueduct. It includes a complex system of archaeological structures spanning various periods and typologies: the homonymous early Christian basilica, built between the 1st and 2nd centuries AD but completed only after the Baroque restorations in the mid-18th century; the Circo Variano, also from the 1st century AD, discovered and identified only through excavations in the 20th century; the Amphitheatrum Castrense, probably built by Elagabalus in the 2nd–3rd centuries AD on the highest point of the area; the Domus Costantiniane, with imperial-era masonry structures excavated during mid-20th century works; and the Civil Basilica of the Sessorian Palace, also known as the Temple of Venus and Cupid (circa 4th century AD) [21].

The project was conducted in collaboration with and with the permission of the Ministry of Culture (MiC) and the *Soprintendenza Speciale di Roma Archeologia Belle Arti Paesaggio*, aiming not only at an in-depth analysis of the archaeological area of Santa Croce but primarily at the development of an interactive and immersive experience in virtual reality. This is an experience to allow future users to explore the evolution of these places in an alternative way, exploiting the active and emotional nature of a multisensory simulation. An integrated survey was therefore necessary to create a digital twin of the site, an accurate replica of reality used as a realistic digital environment for a freely explorable virtual museum. This exhibition would be characterized by forms and behaviors intentionally discordant with the actual location, blending high-tech and metaphysical elements.

The laser scanning survey was planned according to a series of different criteria. The scanning was conducted in two phases during September 2022 (Figure 17), using the morning hours to achieve uniform lighting. In the first phase, special attention was given to scanning the Claudian Aqueduct, the Aurelian Walls, and the archaeological remains of the Circo Variano, as these elements were essential to the project; therefore, a high level of detail was necessary for their scanning. In the second phase, the scanning initially focused on the Domus Costantiniane, including a section of the adjacent aqueduct. Subsequently, the scanning campaign proceeded to the area beyond the tree-lined path, intended to serve as a background

for the project. However, a lower level of detail was chosen to avoid excessive data overload. The laser scanner used for the survey was the Z+F IMAGER 5010C.

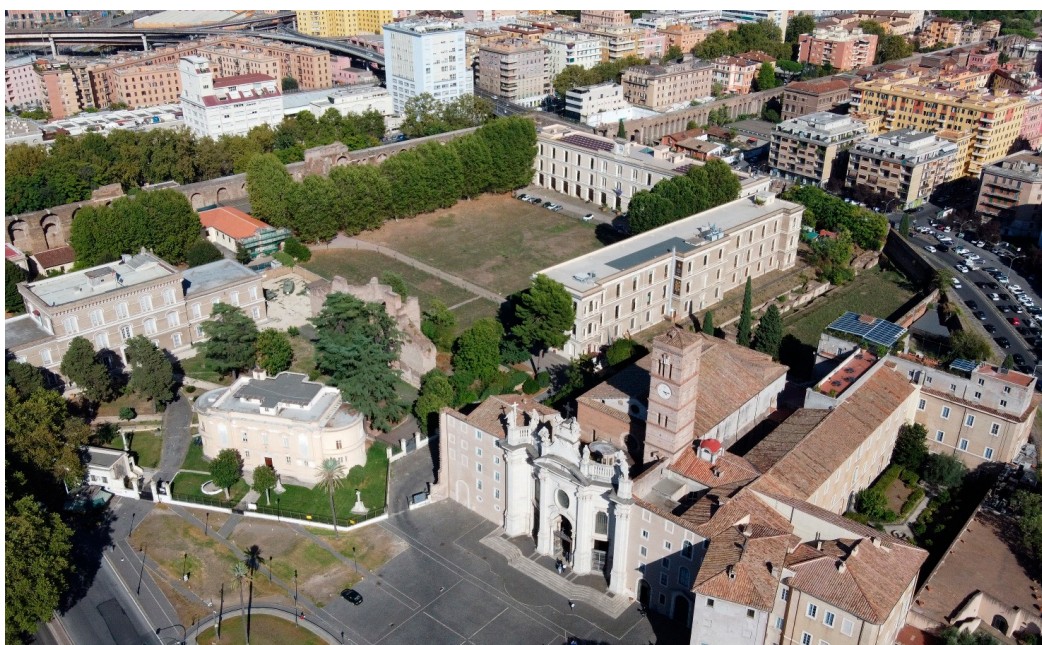

**Figure 16.** Aerial image of the Santa Croce in the Gerusalemme site, captured during the survey campaign. The area is particularly complex due to the historical–archaeological stratifications that distinguish the Horti Variani area: the Claudian Aqueduct, the Aurelian Walls, the archaeological remains of the Circo Variano, the various Domus Costantiniane, the church and the annexed Amphitheatrum Castrense, the cupid temple, and more (image by the authors).

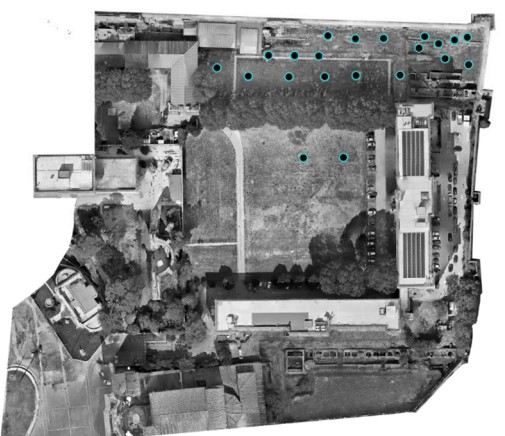
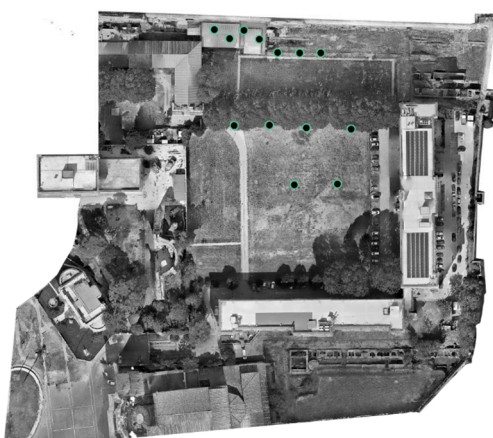

**Figure 17.** Plan of the first and second phases of the laser scanner survey campaign. The survey project was set up to avoid possible blind areas due to the arches of the aqueduct and the numerous wall ruins which could occlude the laser scanner. Therefore, stations were programmed on paper, calibrated on site where necessary, and all performed at a high resolution, to create a dense point cloud capable of expressing the material details of the survey (image by the authors).

Initial operations included tool calibration, setting the resolution, scanning speed, and accuracy. Survey points were then taken along the defined internal and external paths of the designed campaign. Special attention was given to collimation points between interconnected spaces that were not completely visible, ensuring control over possible blind areas of the point cloud. After completing all the scans for both phases, the two point clouds were generated and merged using Autodesk Recap 2022. The merging was achieved

by recognizing common points between the models within the project, resulting in a unified dense point cloud. The result included a total of 30 scans from various positions within the Santa Croce in the Gerusalemme archaeological area. Due to the high level of detail used during the first phase, the resulting point cloud was extremely heavy, causing file management issues. Therefore, it was decided to optimize the number of points to reduce its size. However, a different approach was taken for the area of the domus and the aqueduct, whose number of points remained high to ensure greater accuracy. Through this process, defective areas were identified in the coloring of the point cloud, caused by the interference of sunlight during data acquisition. However, this issue was anticipated from the beginning, and a second survey with a drone was planned to integrate panoramic data acquired by the laser scanner.

Once the generation of the dense point cloud obtained from the laser scanner was completed, the model was exported in E57 format and subsequently imported into the structure-from-motion software Agisoft Metashape Professional 1.7.2. This process was necessary to divide the dense point cloud into four distinct chunks using selection and point removal tools. This division was carried out to optimize operations and set each chunk specifically based on the role the scanned subject would have within the project, considering the final goal. The four chunks corresponded to different areas: the Domus Costantiniane, the Claudian Aqueduct, the Aurelian Walls and Circo Variano, and the urban context.

The survey by UAV, planned in collaboration with an external certified technician, was also divided into two phases (Figure 18). During the first phase, the focus was on the Claudian Aqueduct and the Aurelian Walls, fundamental for the project and therefore captured with a high level of detail. The second phase was dedicated to a survey of the domus, protected by a shelter about two meters tall, which created some flying difficulties due to the limited space; the operation was nonetheless successfully completed. Subsequently, the surrounding urban context of the archaeological area of Santa Croce in Gerusalemme was scanned to be used as a background for the virtual experience. To initiate the survey process on the Claudian Aqueduct and the Aurelian Walls, targets were positioned and later used as reference points for the composition and processing of the point cloud using Agisoft Metashape. Different types of drones were used as surveying tools, including the DJI Mavic 2 Pro (Claudian Aqueduct), Mavic 3 Cinema (Claudian Aqueduct, Aurelian Walls e Circo Variano, Background), and DJI Mini 2 (Domus, Claudian Aqueduct, Background). For example, the DJI Mini 2, smaller and lighter than the others but equipped with a good camera, was used for the photogrammetric survey of the Domus Costantiniane, covered by a shelter, which required special attention in piloting to avoid damaging the artifacts.

Once the survey phase was completed (Table 1), the process of importing and aligning the images acquired through UAVs was carried out using the SfM software Agisoft Metashape Professional 1.7.2. A sparse point cloud was generated first, followed by the creation of a highly dense point cloud. Afterwards, the same process of division of the point cloud into four chunks was performed, similar to what was done for the one derived from the laser scanner. This operation was carried out with special attention to ensure that the chunks were delimited the same way as those previously obtained.

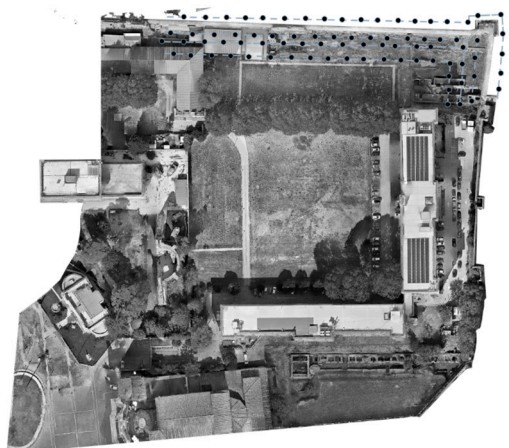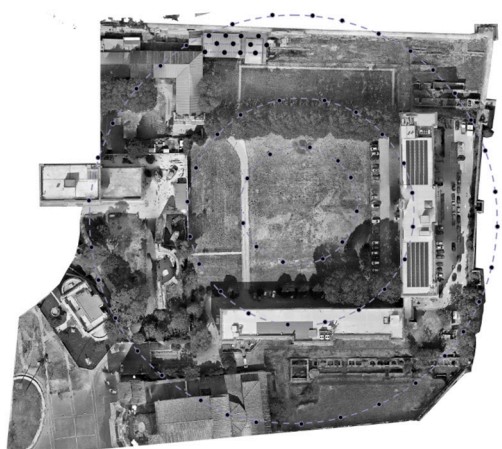

**Figure 18.** Plan of the first and second phase of the aerial photogrammetric survey campaign by drone. Acquisition of the photos was accomplished following the two main shooting techniques for reality-based models: with parallel axes and with converging axes. For some areas, such as for the Domus Costantiniane, hidden by an opaque roof, a precision flight was performed manually to avoid obstacles like supporting pillars (image by the authors).

**Table 1.** Summary of survey parameters and output data, divided according to the chunks.

| | Domus. Point Cloud Laser Scanner/Drone + Texture Drone | Claudian Aqueduct. Point Cloud Laser Scanner/Drone + Texture Drone | | | Aurelian Walls + Circo Variano. Point Cloud Laser Scanner/Drone + Texture Drone | Background. Point Cloud Drone + Texture Drone | |
|---|---|---|---|---|---|---|---|
| **Drone** | DJI Mini 2 | DJI Mavic 3 Cine | DJI Mavic 2 Pro | DJI Mini 2 | DJI Mavic 3 Cine | DJI Mavic 3 Cine | DJI Mini 2 |
| **Images** | 21 | 556 | 4 | 21 | 190 | 643 | 5 |
| **Camera** | FC7303 | L2D-20c | L1D-20c | FC7303 | L2D-20c | L2D-20c | FC7303 |
| **Resolution** | 4000 × 2250 pixel | 5280 × 3956 pixel | 5472 × 3648 pixel | 4000 × 2250 pixel | 5280 × 3956 pixel | 5280 × 3956 pixel | 4000 × 2250 pixel |
| **Focal length** | 4.49 mm | 12.29 mm | 10.26 mm | 4.49 mm | 12.29 mm | 12.29 mm | 4.49 mm |
| **Pixel size** | 1.76 × 1.76 μm | 3.36 × 3.36 μm | 2.41 × 2.41 μm | 1.76 × 1.76 μm | 3.36 × 3.36 μm | 3.36 × 3.36 μm | 1.76 × 1.76 μm |
| **Aligned cameras** | 578 | 552 | | | 190 | 648 | |
| **Coordinate system** | WGS 84 (EPSG::4326). | WGS 84 (EPSG::4326). | | | WGS 84 (EPSG::4326). | WGS 84 (EPSG::4326). | |
| **Point cloud** | 445,755 points | 180,164 points | | | 107,946 points | 435,376 points | |
| **Accuracy** | Highest | High | | | Highest | Highest | |
| **Matching time** | 10 min 28 s | 7 min 12 s | | | 1 min 42 s | 5 min 57 s | |
| **Matching memory usage** | 670.27 MB | 2.55 GB | | | 957.68 MB | 2.94 GB | |
| **Alignment time** | 12 min 46 s | 11 min 30 s | | | 3 min 27 s | 20 min 18 s | |
| **Alignment memory usage** | 271.74 MB | 283.58 MB | | | 93.32 MB | 520.86 MB | |
| **File size** | 53.60 MB | 41.35 MB | | | 18.85 MB | 63.85 MB | |
| **Dense point cloud** | 19,534,249 points | 142,211,824 points | | | 78,761,296 points | 198,642,533 points | |
| **Processing time** | 20 min 32 s | 53 min 25 s | | | 5 h 32 min | 2 h 25 min | |
| **File size** | 300.82 MB | 2.09 GB | | | 3.80 GB | 2.66 GB | |
| **Faces** | 43,323,442 | 10,535,455 | | | 24,999,999 | 4,075,419 | |
| **Vertices** | 21,716,129 | 5,281,503 | | | 17,030,491 | 4,370,785 | |
| **Texture size** | 12.288 × 12.88 × 4 | 4.096 × 4.096 × 12 | | | 4.096 × 4.096 × 4 | 4.096 × 4.096 × 4 | |
| **Reconstruction time** | 1 h 29 min | 4 min 8 s | | | 4 h 2 min | 1 h 38 min | |
| **Uv mapping time** | 28 min 51 s | 7 min 57 s | | | 2 h 6 min | 26 min 42 s | |
| **Uv mapping memory usage** | 9.81 GB | 3.82 GB | | | 3.91 GB | 8.75 GB | |
| **Blending time** | 11 min 18 s | 17 min 17 s | | | 2 min 22 s | 9 min 26 s | |
| **Blending memory usage** | 23.99 GB | 14.97 GB | | | 6.82 GB | 21.41 GB | |
| **File size** | 2.65 GB | 681.17 MB | | | 1.23 GB | 128.88 MB | |

This provides an overview of the settings used for individual monuments, as well as the quality of the outputs developed, such as point clouds and meshes (image by the authors).

A careful evaluation of the data obtained from both campaigns indicated some key considerations for outlining an effective methodology for the following model development and texturing. The remarkable accuracy of the data acquired during the laser scanner survey allowed the generation of a dense point cloud that was very true to reality; however, the use of only panoramic photos proved to be less effective for a high-quality reconstruction of the textures of the mesh model. On the other hand, the use of aerial photogrammetry by UAVs made it possible to obtain more detailed and precise textural information, despite a sparser point cloud compared with the laser scanner. The reconstruction algorithm in the software, working with many shots taken from various positions, selected portions of the photos that best approximated a direction perpendicular to the polygon normal. These portions were combined and reprojected onto the modeled object as an image mosaic, generating the final texture. Therefore, the decision was to use the point cloud from the laser scanner to generate the mesh and then apply the texture developed through the drone survey.

The two dense point clouds from the distinct surveys were processed separately using Agisoft Metashape and later exported in E57 format. A unique reference coordinate system (RDN2008/Italy zone (N-E)—EPSG: 6875) was chosen to align them using the CloudCompare software, given the absence of georeferencing in the laser scanner data compared with the data acquired from the drone (Figure 19).

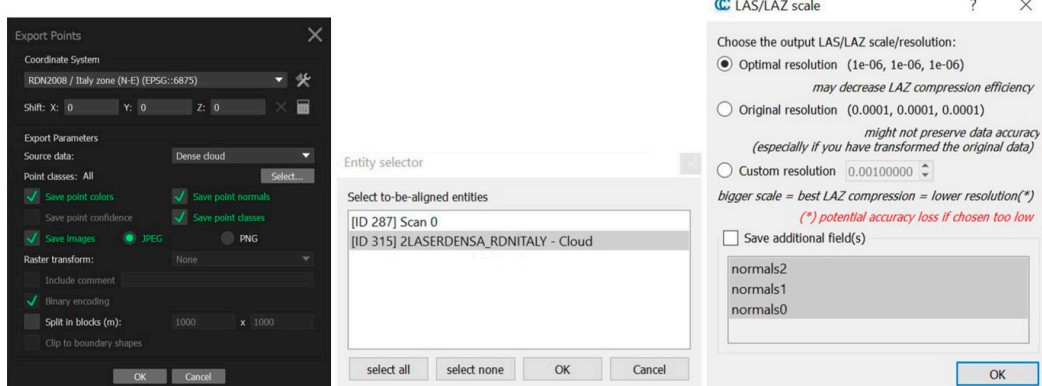

**Figure 19.** From left: export settings screen for the dense point clouds from laser scanners and drones in Agisoft Metashape; import screens in CloudCompare for the previously exported files (image by the authors).

Once the data was imported into the software, a set of matching pairs of points was identified between the two point clouds, which was used for alignment. The model obtained from the drone was used as the reference base for aligning the one from the laser scanner. The selection of common points between the clouds was performed with great care, as the precision in defining them directly affected the coherence of the overlap between the dense point clouds. The goal was to keep the displacement error between the two models constantly below 0.01 m to obtain an accurate alignment, finally achieving a value of 0.001 m (Figure 20).

The software provided the option to adjust the scale between the point clouds during the alignment process (Figure 21), transforming the second point cloud based on the primary one to achieve a perfect fit. In this operation, the larger the displacement error, the more significant the scale adjustment. Once aligned, the two point clouds were kept in two manageable chunks. Subsequently, the aligned point clouds were exported again in LAS format, using an optimal resolution from CloudCompare and preserving the new coordinates.

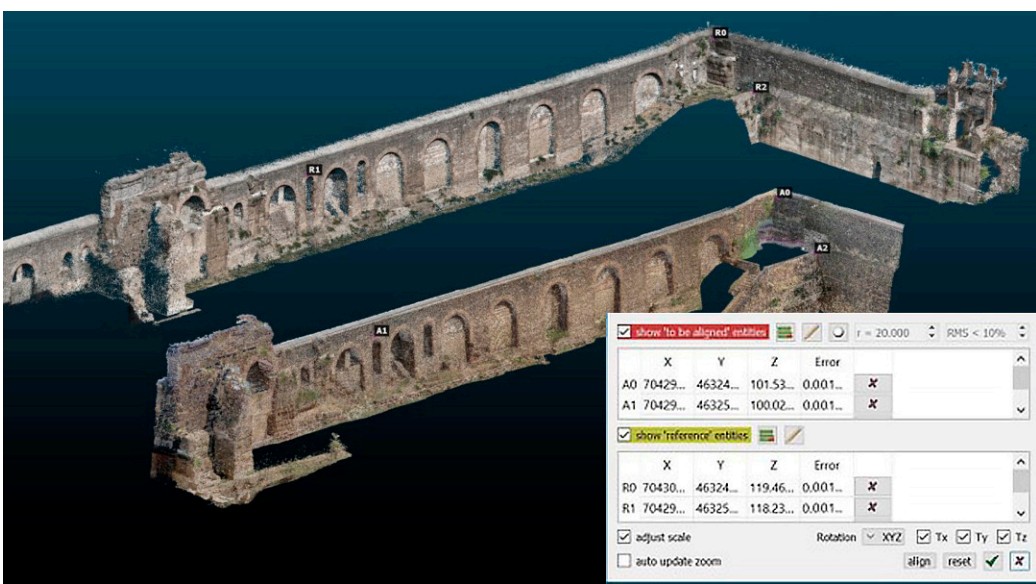

**Figure 20.** Selection of common points between the two imported point clouds in CloudCompare. The choice of targets for the alignment of the two point clouds must be such as to guarantee some characteristics: the points must be rather distant, to avoid a specific scaling and rototranslation in favor of a general one (in this case, for example, the length of the aqueduct); the points must belong to at least two different and/or orthogonal planes, preferably on the three Cartesian planes (image by the authors).

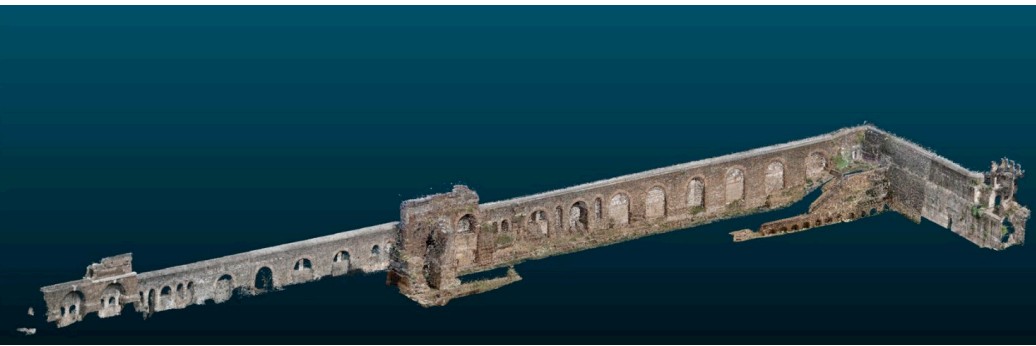

**Figure 21.** Aligned point clouds from drone survey and laser scanning in CloudCompare. The overlap error between the two clouds is 0.001 m, optimal for proceeding to the next stage by returning to the Metashape software (image by the authors).

The next step was to reintroduce the two aligned point clouds into Agisoft Metashape, divided into two separate chunks, to proceed with the construction of the mesh model. During the meshing phase, the laser scanner point cloud was processed with a high level of detail, disabling the "face count" option to avoid potential reductions in the number of surface polygons (Figure 22).

The obtained mesh was then subjected to the texturing phase using photos captured by the drone, applying the "Mosaic" blending mode to obtain the final high-resolution JPEG texture (12,288 × 12,228 pixels). This process, from the alignment phase in CloudCompare to the texturing phase in Agisoft Metashape, was replicated for each of the four areas that constituted the entire project. This resulted in the creation of four distinct files, each configured specifically based on the expected final output (Figure 23).

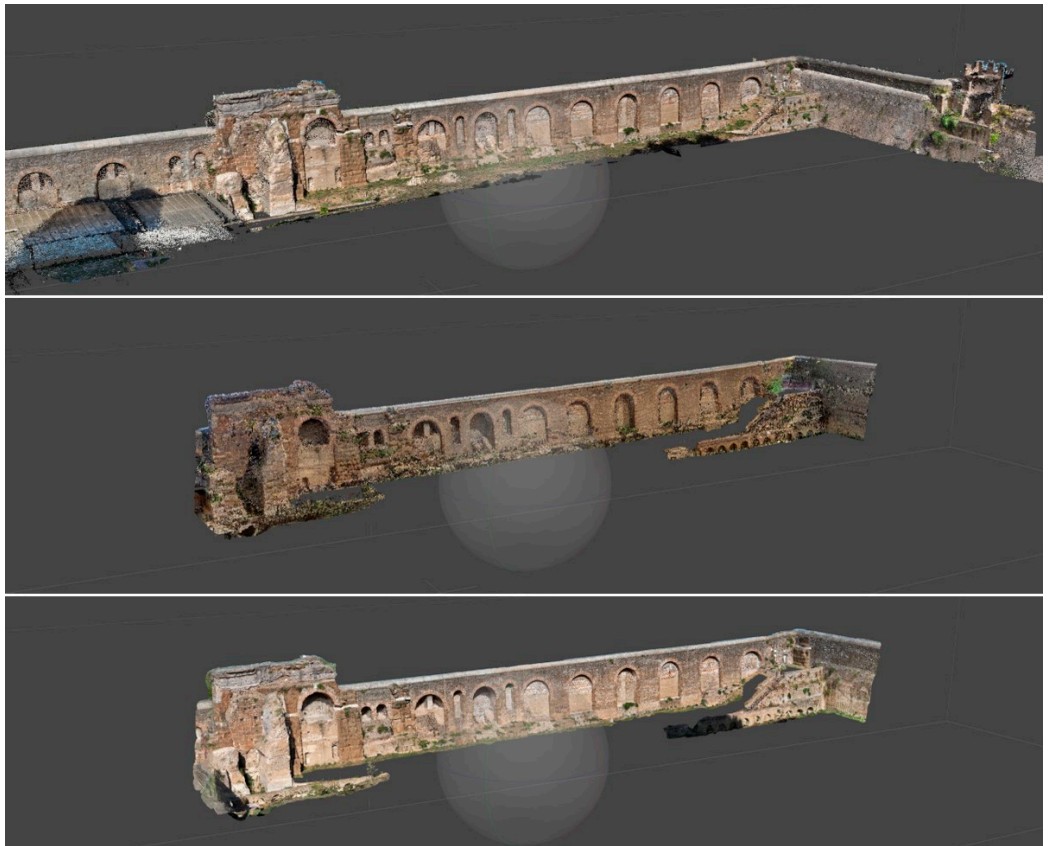

**Figure 22.** The dense laser scanner point cloud was exported from CloudCompare with the new Cartesian position coincident with the lower-quality drone point cloud. This way, it was possible to import the laser cloud into the Metashape source file, in the same position, and proceed to the creation of a mesh model more detailed than one from SfM. The 3D model finally underwent the texturing process, exploiting the photos taken by drone. From the top: dense point cloud from drone survey; dense point cloud from laser scanning; laser-scanner-derived mesh, textured by the drone survey (images by the authors).

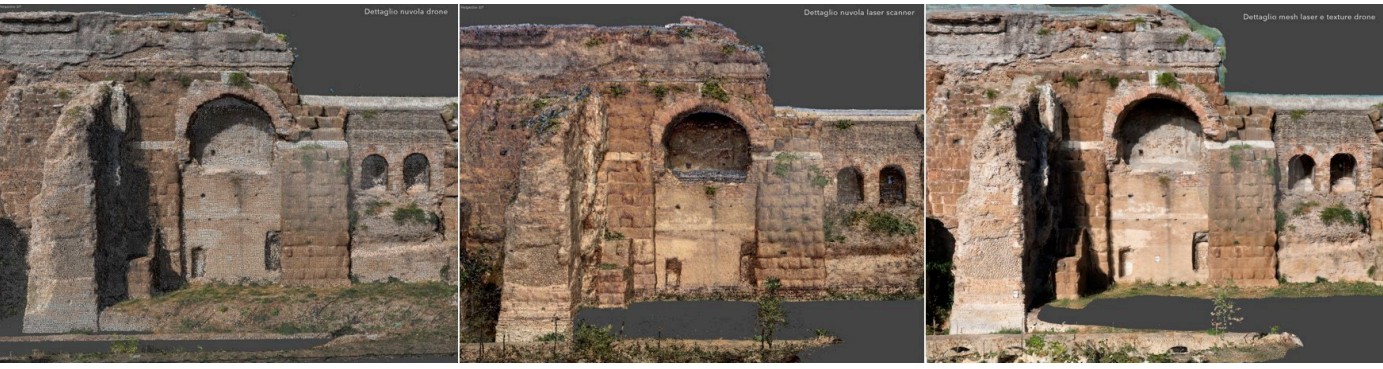

**Figure 23.** From left to right: focus on the dense point cloud from drone survey; focus on the dense point cloud from laser scanning; focus on the mesh from laser scanning textured by the drone survey. The number of points in the laser scanner cloud made it possible to mesh with a high resolution. The perfect coincidence of the two models, laser and drone, has allowed the creation of a high-resolution texture from the set of photographs derived from the flights. This allowed for the creation of a coherent digital twin close to reality, on which the immersive VR experience could be based (images by the authors).

Once the processes of meshing and texturing for the entire archaeological area included in the project were completed, it was necessary to further divide each file into four chunks to achieve a proper export in OBJ format to be imported into Unreal Engine 5.0.3, the software chosen to develop the virtual experience. This procedure was carried out with the aim of optimizing the graphics performance of the project in virtual reality. Considering that the maximum size limit for importing into UE5 is around 1.5 GB per OBJ file, numerous exports were made from Agisoft Metashape, keeping each group of meshes below this threshold. Furthermore, each exported mesh group was divided to have a small overlapping area with adjacent ones, to ensure a correct overlap when placed into UE5 environment, avoiding gaps in the final model. During the export process, the same local reference system was consistently used, devoid of georeferencing, as the final project output did not require such information. Given that the overall project included four distinct files, four different local reference systems were employed. Once imported into Unreal Engine, this resulted in a different arrangement of the archaeological artifacts compared with their real-world localization. To facilitate the correct placement of the parts, a complete version of the archaeological area was also included as a reference, to be then removed at the end of the process. This complete version was obtained by meshing the drone-generated point cloud at a lower quality to keep its weight below 1.5 GB. At this point, the original positions of individual archaeological artifacts in high quality were recomposed by adjusting the position and rotation coordinates of the parts.

The development of the VR experience on UE5 was conducted on a workstation equipped with a Windows 10 Pro 64-bit OS, 48 GB of RAM, an Intel(R) Core(TM) i7-8700 CPU, and a NVIDIA GeForce RTX 3090 Ti GPU. In order to handle the highly detailed meshes produced earlier, which had a high poly count, these were imported while activating the dedicated Nanite geometry visualization system (Figure 24). Recently developed and implemented in Unreal Engine 5, Nanite allows the real-time rendering of elements with a very high object count based on the observer's distance, allowing continuous internal optimization without necessarily simplifying assets in advance through other programs. Nanite is crucial for using digital twins with high levels of detail (LOD), especially when combined in a complex scene like Santa Croce in Gerusalemme [22,23]. Otherwise, hardware resources would struggle to manage the enormous amount of data. The Nanite mesh essentially remains a triangle mesh, which is automatically analyzed and segmented into clusters of triangles. These clusters dynamically adjust their LOD, increasing or decreasing it based on the camera's position, allocating more memory to visible details [24]. Nanite does not support objects without opaque and single-sided materials yet, but this is not an issue for the imported digital twin, as opaque and structurally simple materials were applied to the surfaces.

The VR Museum experience of the archaeological site was developed using a standard VR template provided by the game engine. This template was specifically designed to simplify the initial scene setup and already included useful scene elements, coding components, interactions, and animations. The entire experience was composed of three interconnected scenarios, created by blending scanned artifacts, hand-modeled elements in Rhinoceros 7, and high-resolution objects and materials sourced directly from the Megascans library by Quixel, accessible within the UE5 editor. The library facilitated the employment of digital vegetation to enhance the environments, chosen according to the species existing at the actual sites. The main scene, representing the archaeological area, was enriched with abstract designed artifacts to guide the user's experience. Additionally, a museum/control room was created within the scene, providing access to other environments (Figure 25).

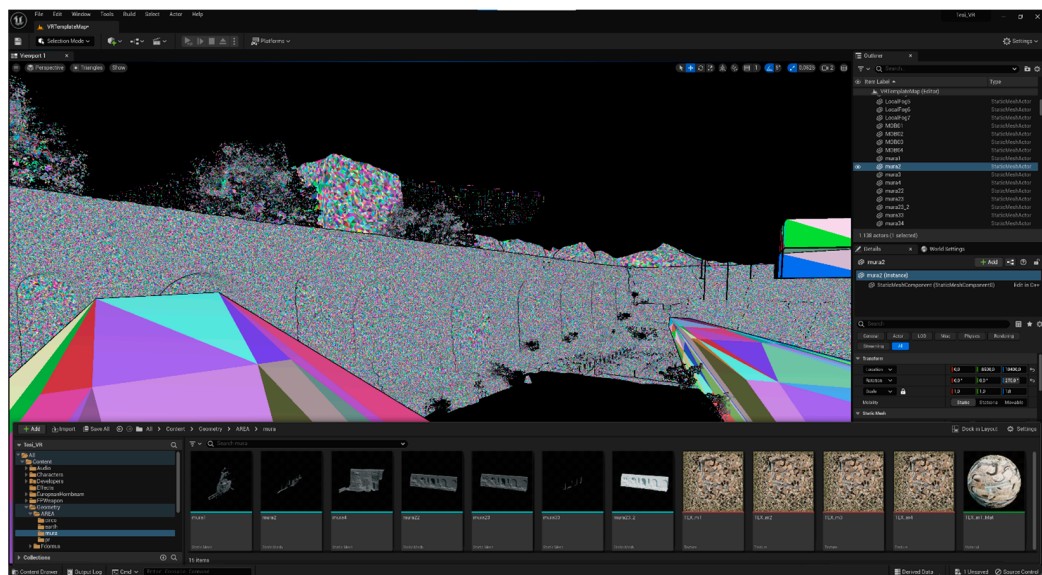

**Figure 24.** Nanite geometry visualization in Unreal Engine 5.0.3: the polygon count of the Aqueduct is visibly higher than for the structure in the background, while the foreground walkways appear much simpler as they were manually modeled and were already less complex (image by the authors).

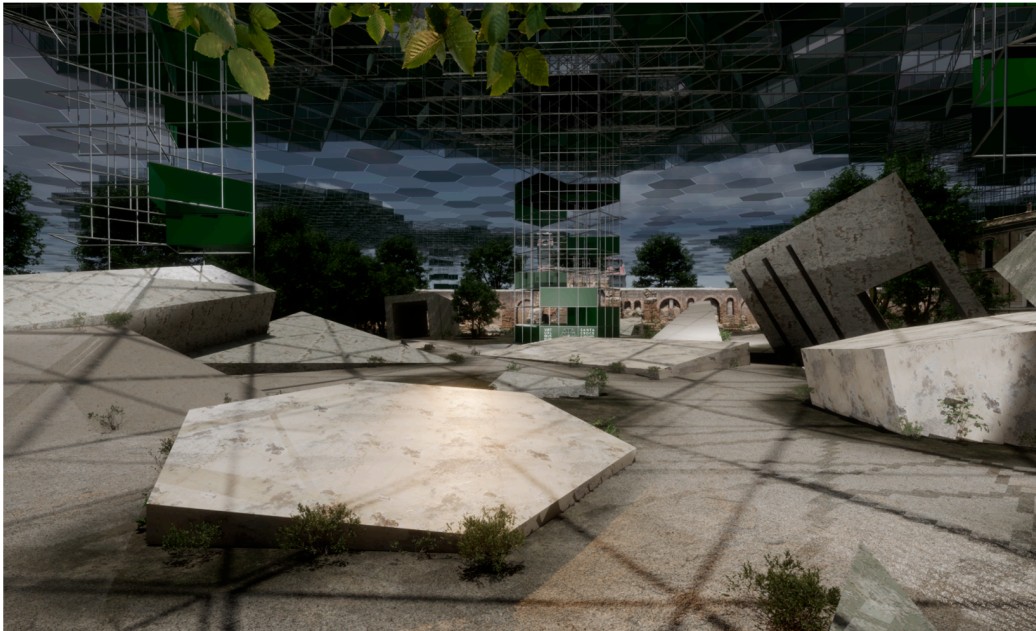

**Figure 25.** A screenshot captured from within the virtual experience of the main scenario. In the background, the digital twin of the Claudian Aqueduct can be seen, while, in the foreground, hand-modeled elements have been placed to create evocative scenery, along with access to the virtual museum (image by the authors).

The other two scenes, the Domus Costantiniane and the Circo Variano, represent two focuses that reuse the meshes produced from the integrated survey to create isolated exhibition rooms. For performance reasons and to create the sensation of a surreal and abstract environment, the transition between scenarios (called maps in UE5) is achieved through a specially coded teleportation system. Other interactive elements, such as popup info points, particle effects, dynamic lights, and music, have been carefully placed in the area and programmed using the internal blueprint coding system to enrich the scenes and make them more engaging for the users. After wearing the headset and exploring a site

which blends reality with metaphysics, they can discover and learn about the history of the place (Figures 26 and 27).

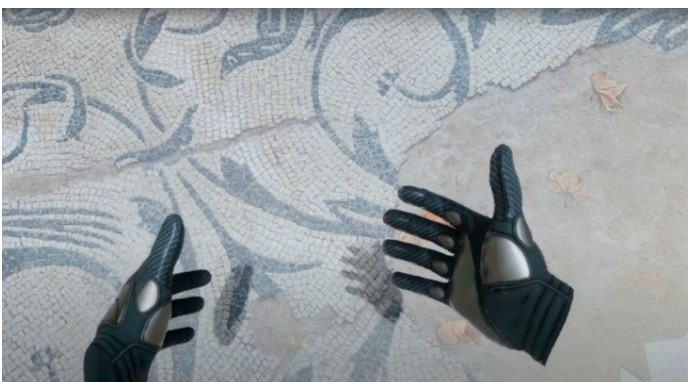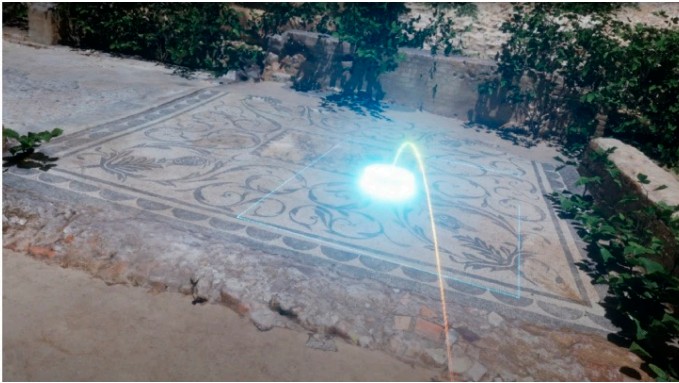

**Figure 26.** Screen from within the virtual experience, showing the digital hands the user exploits to interact with the domus (**on the left**) and the teleportation system (**on the right**) that allows movement within the environment (images by the authors).

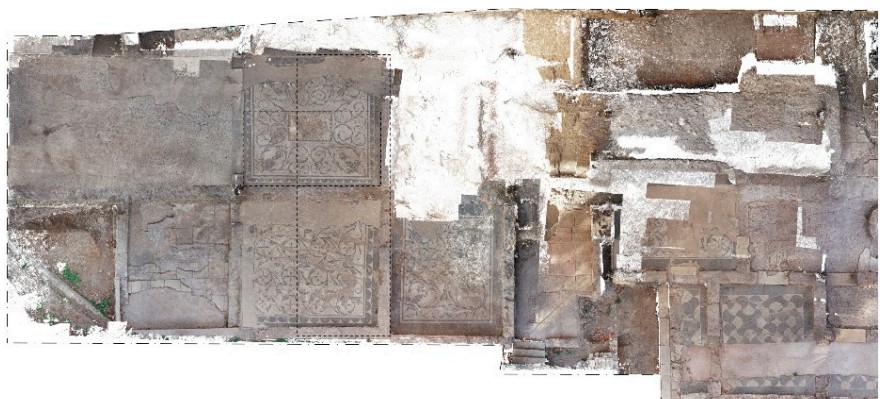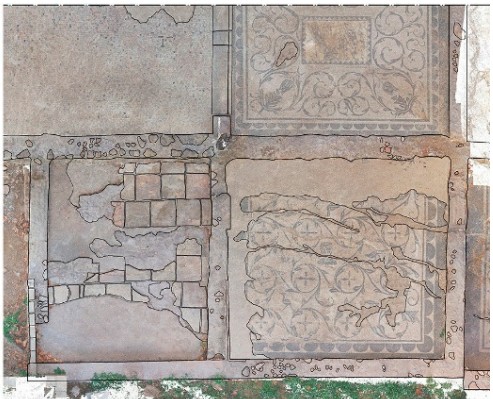

**Figure 27.** Selections of some planimetric elaborations created for the digital exhibition within the virtual museum. On the left, the orthophoto of the domus floors; on the right, a focus on the redrawing of the remaining mosaics (images by the authors).

## 3. Results

Integrated digital surveying through UAV aerial photogrammetry and laser scanning allowed for the creation of highly accurate models of the two research sites, utilizing inherently similar methodologies but adapted for very different purposes. On the one hand, the 2D restitution of Rocca Farnese was carried out for subsequent analysis and restoration interventions, requiring careful consideration of dimensions and materials to accurately represent the current state of the site. On the other hand, the 3D reconstruction of the archeological site of Santa Croce in Gerusalemme aimed at creating an engaging environment in virtual reality, in which mesh surface optimization and integrity were combined with a need for detail and realism to construct an immersive experience of the locations. This reconstruction was also presented at the *Biennale dello Spazio Pubblico 2023*, held by the Department of Architecture, University of Roma Tre, where visitors could immerse themselves in this and other interactive virtual environments using an Oculus Rift S head-mounted display, freely navigating the museum with appropriate controllers and experiencing the archaeological site in an alternative way.

In both case studies, the methodology followed led to the elaboration of point clouds from different surveys, which were successfully combined with a divergence of only 2 mm between the two models for Rocca Farnese (Table 2a) and only 1 mm for those of Santa Croce in Gerusalemme (Table 2b). Once cleaned and refined, the final point clouds allowed

the generation of high-density meshes, which were used without the need for further simplification or retopology due to the ultimate goals of the research and the exploiting of systems such as Nanite.

**Table 2.** Final summary of the results obtained from the models elaborated in the two case studies.

| (a) | | |
|---|---|---|
| **Rocca Farnese** | | |
| **UAV point cloud** | n. points | 181,914,544 |
| **Laser scanner point cloud** | n. points | 638,600,000 |
| **Final point cloud model** | n. points | 820,500,000 |
| | final divergence between joined point clouds (m) | 0.002 |
| **Mesh model** | faces | 39,526,220 |
| | vertices | 19,806,251 |
| (b) | | |
| **Santa Croce in Gerusalemme** | | |
| **UAV point cloud** | n. points | 251,961,595 |
| **Laser scanner point cloud** | n. points | 187,188,307 |
| **Final point cloud model** | n. points | 439,149,902 |
| | final divergence between joined point clouds (m) | 0.001 |
| **Mesh model** | faces | 82,934,315 |
| | vertices | 48,398,908 |

## 4. Discussion

Other than allowing the survey of spaces inaccessible by drone (such as the interiors of the Rocca), the implementation of laser scanning provides greater geometric detail of the areas captured through aerial photogrammetry. This implementation allowed full completion of the architectural volumes of the sites. The use of UAVs also facilitated the reconstruction processes, especially in the case of Rocca Farnese, where dense vegetation and numerous garden levels would have posed further challenges and required more time if attempted only through terrestrial photogrammetry and laser scanning. However, the vegetation, particularly the tree canopies near the artifacts, did affect the acquisition of certain fronts, creating gaps and necessitating the integration of terrestrial data. In the case of Santa Croce in Gerusalemme, tall vegetation did not present the same issue as it was generally situated at a distance from the archaeological site. Low vegetation and shrubs close to the ground connections of the artifacts, however, required meticulous work during mesh cleaning to preserve the captured data. During the VR environment development, efforts were made to fill and conceal any gaps with digital plant assets.

The adoption of an integrated survey methodology, exploiting various indirect techniques, significantly optimized this 2D–3D reconstruction work in architecture and archaeology. It allowed the combination of different levels of detail and addressed obstacles (such as volumetric articulations and vegetation) that are challenging to overcome through direct measurement. The use of orthophotos as a base for reconstructions increased the level of detail in redrawing and streamlined the process compared with traditional redrawing.

The two case studies described here are just examples of possible applications of this methodology. The approach needs to be adapted for cases with different goals, surveyed subjects, and contexts. Although the instruments may be similar and compatible, a survey campaign must always be designed according to the result to be obtained, avoiding waste of energy, resources, and costs. For Rocca Farnese, the client required a two-dimensional survey that would highlight the dimensional and material aspects as the basis for a subsequent map of materials and degradations to plan conservative restoration interventions. In this case, the effort was made to obtain a sufficiently detailed point cloud for CAD redrawing and the creation of strongly detailed orthophotos to ensure clear reading of surfaces. In the case of Santa Croce in Gerusalemme, the purpose of the survey was the

creation of an immersive virtual museum, as established with the *Soprintendenza Speciale di Roma*. This required exploitation of the archaeological remains in the area to create a digital twin with optimal quality for an experience of a virtual space true to reality. Therefore, the density of the laser scanner point cloud (whose textures were low in quality) was exploited to generate a mesh to be integrated with high-resolution textures obtained by UAV photographs.

Nonetheless, integrated surveys may have very different purposes. Many institutions are now digitizing architectural and landscape heritage, creating archives with many goals: from the simple consultation of the asset in a virtual environment, to the analysis or management of the digital copy, to the systematic programming of interventions, up to the analysis of the temporal decay of an asset subjected to many surveys through time. It is possible to structure a survey in HBIM to plan conservation or evaluate possible risks (for example, for buildings damaged by destructive earthquakes). Thanks to the speed of data acquisition, integrated surveys can be exploited to document the progressive stratifications during archaeological excavations, to allow a better understanding of the various covered levels in a virtual environment. In some cases, virtual models deduced from the surveys become the basis for experiences in XR. Especially after the pandemic crisis, museums and institutes are gradually starting to provide virtual tours of portions of their galleries, with various degrees of immersion. [25]. These alternatives have improved digital tourism, especially in a period in which circulation has been prohibited, reaching new frontiers in the creation of interactive digital twins. This process can be followed for those archaeological sites which, for various reasons, are not accessible to the public.

Ultimately, the integrated survey is just a new tool for replicating in a digital environment the same information that was typically represented by analogue survey: architectural or material plans, elevations, and sections in orthogonal projection, which are useful for representations at canonical scales, measurement, and analysis of architecture. These elaborations are still among the most requested by institutions and administrations.

The methodology adopted proved effective in diverse contexts with numerous purposes. While campaigns required various types of integrations during the process, this highlights the importance of an approach that integrates multiple tools and techniques to adapt to the specific needs of each case. The need to balance performance and quality, especially in projects requiring optimization, like those in VR, is an interesting research area that may need further critical exploration, both of the experiences described, and in further research in agreement with the *Soprintendenza Speciale di Roma* or with the *Parco dell'Appia Antica*. Surely, one of the research areas on which the research group is focused is the use of XR to enhance the understanding of places, whether they are real or just designed and virtually reconstructed [26].

The research carried out and the experiments on open conventions allow for the improvement of a process which still has issues, especially related to the significant sizes of files and their management. Therefore, alternative forms of experimentation could deepen the investigation into integrated systems using data, including working on software with innovative algorithms (such as Nanite for Unreal Engine).

From a social and perceptual point of view, VR (or XR in general) is a subject for wider interdisciplinary research, which attempts to study the different effects of experiencing these virtual environments on the personalities and development of new generations. Many sociologists have long criticized the educational power of some role-playing video games, but new studies are taking place on so-called serious games, immersive virtual experiences aimed at educating and teaching. The digital twins created and described in this article are part of these virtual reconstructions, which become navigable environments for the "role player" and which require a realistic and plausible level of detail.

These sectors are now at the center of global media attention, and it is the mission of universities to develop these issues and provide research ideas, investigate solutions, and create interdisciplinary relationships with the aim of improving the reconstruction, management, and perception of digital twins.

**Author Contributions:** Conceptualization, D.C., S.B. and A.C.; methodology, D.C., S.B. and A.C.; software, D.C., S.B. and A.C.; validation, D.C., S.B. and A.C.; formal analysis, D.C., S.B. and A.C.; investigation, D.C., S.B. and A.C.; resources, D.C., S.B. and A.C.; data curation, D.C., S.B. and A.C.; writing—original draft preparation, D.C., S.B. and A.C.; writing—review and editing, D.C., S.B. and A.C.; visualization, D.C., S.B. and A.C.; supervision, D.C.; project administration, D.C.; funding acquisition, D.C. All authors have read and agreed to the published version of the manuscript.

**Funding:** The research concerning the survey of Rocca Farnese was funded by Arch. Anelinda Di Muzio, Ph.D, specialist in monument restoration, on behalf of the *Soprintendenza Archeologia Belle Arti e Paesaggio per la Provincia di Viterbo e per l'Etruria Meridionale*, by scientific collaboration agreement *Analisi, dimensionamento e restituzione materica delle superfici della Rocca Farnese a Capodimonte (VT) attraverso tecnologie avanzate di misurazione e rappresentazione*—Dipartimento di Architettura Università Roma Tre.

**Data Availability Statement:** All data included in this paper is strictly connected to the research, is not yet published, and is the result of personal studies and analysis carried out by the research group.

**Acknowledgments:** This research was made possible by collaboration with Arch. Anelinda Di Muzio, *Soprintendenza Archeologia Belle Arti e Paesaggio per la Provincia di Viterbo e per l'Etruria Meridionale*, and Alessandro D'Accolti for Rocca Farnese, and with *Ministero della Cultura* (MiC), Arch. Alessandra Centroni, and Dott.ssa Simona Morretta on behalf of the *Soprintendenza Speciale di Roma Archeologia Belle Arti Paesaggio* for the archeological site of Santa Croce in Gerusalemme.

**Conflicts of Interest:** The authors declare no conflict of interest.

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
