# Peer review of "Integrated Surveying, from Laser Scanning to UAV Systems, for Detailed Documentation of Architectural and Archeological Heritage"

_drones, doi:10.3390/drones7090568_

Round 1

Reviewer 1 Report

The submitted paper presents two case studies that integrate spatial data from laser scanners and photogrammetry. The topic  and the approach are not innovative, however some difficulties to integrate these techniques persist and are addressed in this work. 

The presentation is well organised, but the paper would benefit with a deeper discussion of the  results, and the accomplished integration of spatial data from laser scanners and photogrammetry into CAD systems. 

Moreover, a systematic presentation about the precision of the data might be also appreciated. 

ln 66: "... photogrammetric captures must occur in sequences of 64 shots with uniform characteristics, depicting the object with a certain overlap percentage 65 (at least 30%) to allow for the recognition of homologous points within the set of images." 30% could be insufficient, I recommend authors to include a reference to support or improve the statement.

ln 567: "The goal was to keep the displacement error between the two models always below 0.01 to obtain an accurate alignment, and finally archiving a value of 0.001 (Figure 20)."  References to units are missing. The legibility of fig. 20 need improvement.

The text seems  well written.

Author Response

REVIEWER 1

The presentation is well organised, but the paper would benefit with a deeper discussion of the results, and the accomplished integration of spatial data from laser scanners and photogrammetry into CAD systems. 

Moreover, a systematic presentation about the precision of the data might be also appreciated. 

The results were implemented and integrated with a final table underlining the main data obtained by the elaboration of point clouds and mesh models through the followed methodology (lines 766-773).

ln 66: "... photogrammetric captures must occur in sequences of 64 shots with uniform characteristics, depicting the object with a certain overlap percentage 65 (at least 30%) to allow for the recognition of homologous points within the set of images." 30% could be insufficient, I recommend authors to include a reference to support or improve the statement.

We improved the statement and included new references to support it, and to implement the introduction as highlighted (references changed their own numbers consequently). (line 67)

ln 567: "The goal was to keep the displacement error between the two models aalways below 0.01 to obtain an accurate alignment, and finally archiving a value of 0.001 (Figure 20)."  References to units are missing. The legibility of fig. 20 need improvement.

Units were added and fig. 20 was modified to improve the legibility, especially of the table values. (line 625-627)

Reviewer 2 Report

This paper presents a reasonable method to solve a real application problem. It is well-organized, clearly written, and shows some interesting results that encouraged it to be accepted with major revision. However, the commented questions need only to be answered.

1. The introduction and related work are mixed into a long section. It would be more clearly for readers to separate them.
2.    More studies of archeological heritage documentation should be cited, e.g. 1) A 3D Spatial Diagnostic Framework of Sustainable Historic and Cultural District Preservation: A Case Study in Henan, China, Buildings, 2023.  2) Applying Geomatics Techniques for Documenting Heritage Buildings in Aswan Region, Egypt: A Case Study of the Temple of Abu Simbel, Heritage, 2023. 3) Documentation and Virtualisation of Vernacular Cultural Heritage: The Case of Underground Wine Cellars in Atauta (Soria), Heritage, 2023. 4) Application of 3D Laser Scanning Technology Using Laser Radar System to Error Analysis in the Curtain Wall Construction, Remote Sensing, 2022. I cannot summarize all of them, but the authors are expected to reorganize these studies.
3.     Fig.2, Fig.8, Fig 9, and Fig.11 are not clearly shown.
4.    Please enrich the captions of all figures and tables for clarification.
5.    It is better to write Table 1 in the latex code or office not to put a picture of it because the table is not clearly shown.
 6. When the author(s) refer to their 3D modeling, do they in fact mean the conservation or restoration of the monument? 
7. I assume there are items authors wish to document and report on the paper?
8.  Were the points all taken with the GNSS or did you add extra control with only the Total Station?
 9. How was the point cloud transferred and what architectural software was used?
10.  Are these Total Station control points?
11. In order to process the laser scanning data, are the scans removed unwanted or scattered points within each individual or combined scan?
12. Why were there only 103 scans needed for the interior space, one-line explanation would probably suffice here.

13.    I also find some grammar problems in this paper. The author needs to carefully check these low mistakes, which is very important for readers.

This paper presents a reasonable method to solve a real application problem. It is well-organized, clearly written, and shows some interesting results that encouraged it to be accepted with major revision. However, the commented questions need only to be answered.

1. The introduction and related work are mixed into a long section. It would be more clearly for readers to separate them.
2.    More studies of archeological heritage documentation should be cited, e.g. 1) A 3D Spatial Diagnostic Framework of Sustainable Historic and Cultural District Preservation: A Case Study in Henan, China, Buildings, 2023.  2) Applying Geomatics Techniques for Documenting Heritage Buildings in Aswan Region, Egypt: A Case Study of the Temple of Abu Simbel, Heritage, 2023. 3) Documentation and Virtualisation of Vernacular Cultural Heritage: The Case of Underground Wine Cellars in Atauta (Soria), Heritage, 2023. 4) Application of 3D Laser Scanning Technology Using Laser Radar System to Error Analysis in the Curtain Wall Construction, Remote Sensing, 2022. I cannot summarize all of them, but the authors are expected to reorganize these studies.
3.     Fig.2, Fig.8, Fig 9, and Fig.11 are not clearly shown.
4.    Please enrich the captions of all figures and tables for clarification.
5.    It is better to write Table 1 in the latex code or office not to put a picture of it because the table is not clearly shown.
 6. When the author(s) refer to their 3D modeling, do they in fact mean the conservation or restoration of the monument? 
7. I assume there are items authors wish to document and report on the paper?
8.  Were the points all taken with the GNSS or did you add extra control with only the Total Station?
 9. How was the point cloud transferred and what architectural software was used?
10.  Are these Total Station control points?
11. In order to process the laser scanning data, are the scans removed unwanted or scattered points within each individual or combined scan?
12. Why were there only 103 scans needed for the interior space, one-line explanation would probably suffice here.
13.    I also find some grammar problems in this paper. The author needs to carefully check these low mistakes, which is very important for readers.

Author Response

REVIEWER 2

This paper presents a reasonable method to solve a real application problem. It is well-organized, clearly written, and shows some interesting results that encouraged it to be accepted with major revision. However, the commented questions need only to be answered.

  1. The introduction and related work are mixed into a long section. It would be more clearly for readers to separate them.

Though we understand the need given by this comment, the subdivision rules of the template don’t allow for a separation of introduction into more sections.

  1. More studies of archeological heritage documentation should be cited, e.g. 1) A 3D Spatial Diagnostic Framework of Sustainable Historic and Cultural District Preservation: A Case Study in Henan, China, Buildings, 2023. 2) Applying Geomatics Techniques for Documenting Heritage Buildings in Aswan Region, Egypt: A Case Study of the Temple of Abu Simbel, Heritage, 2023. 3) Documentation and Virtualisation of Vernacular Cultural Heritage: The Case of Underground Wine Cellars in Atauta (Soria), Heritage, 2023. 4) Application of 3D Laser Scanning Technology Using Laser Radar System to Error Analysis in the Curtain Wall Construction, Remote Sensing, 2022. I cannot summarize all of them, but the authors are expected to reorganize these studies.

We decided to cite the suggested cases to enrich the references to cases related to the work presented in the paper, underlining how similar approaches can answer different goals (lines 154-166)

3Fig.2, Fig.8, Fig 9, and Fig.11 are not clearly shown.

We verified the resolution of the figures: figs. 8-9-11 (attached as separates files as the rest of images) are clear and have the right resolution as requested. As for the fig. 2, we gave it with the correct resolution, but the quality given by the original source was low (these boards have also a wide dimension and texts couldn’t be clear in a smaller reproduction anyways).

  1. Please enrich the captions of all figures and tables for clarification.

We enriched the captions were needed to better explain figures and tables.

  1. It is better to write Table 1 in the latex code or office not to put a picture of it because the table is not clearly shown.

Table 1 was replaced with an office table instead of a figure.

  1. When the author(s) refer to their 3D modeling, do they in fact mean the conservation or restoration of the monument?

The two case studies presented exploited 3D modelling for two different goals (as it’s highlighted in lines 177-181 and later in section 2), where for Rocca Farnese 3D modelling had aimed at supporting the restoration of part of the monument, while for Santa Croce in Gerusalemme the purpose was the dissemination of the archeological area. While it was not the main goal, these models can also provide support for conservation operations (as we decided to specify in lines 154-166, answering point 2 of this review).

  1. I assume there are items authors wish to document and report on the paper?

We didn’t understand which items the reviewer is referring to.

  1. Were the points all taken with the GNSS or did you add extra control with only the Total Station?

Target points’ coordinates were taken by a total station connected to a GNSS. We added a text on line 231 and also in the captions of fig. 8 and fig.9.

  1. How was the point cloud transferred and what architectural software was used?

For Rocca Farnese, the UAV point cloud was transferred from Agistoft Metashape Professional 1.7.2 to Autodesk ReCAP 2023 by exporting and importing it as an E57 file (lines 367-370). The project made in Autodesk ReCAP 2023 (lines 369-370) was then directly imported in AutoCAD 2022 (which was the architectural software used) as an externa reference to our drawings (RCP file, lines 384-388). For Santa Croce in Gerusalemme, the point cloud acquired by laser scanner was elaborated in ReCAP and then transferred as a E57 file into Metashape (lines 539-541), which was the same extension used for the point cloud acquired by drone when both of them were imported into Cloud Compare (lines 611-615), and later transferred again on Metashape as a LAS file (lines 639-648). From here on, the model was meshed and transferred as an OBJ file to be used on Unreal Engine 5.0.3. (lines 675-678).

  1. Are these Total Station control points?

Yes they are. As referred on line 231 in comment 8

  1. In order to process the laser scanning data, are the scans removed unwanted or scattered points within each individual or combined scan?

We specified the cleaning process also adding a phrase in lines 325-327

  1. Why were there only 103 scans needed for the interior space, one-line explanation would probably suffice here.

We only needed to survey the ground floor (as it was first underlined in line 236 and also later), and the rooms were quite small (as it was explicated in lines 247-249).

  1. I also find some grammar problems in this paper. The author needs to carefully check these low mistakes, which is very important for readers.

We read all the text and corrected some grammar errors we found out.

Reviewer 3 Report

This paper describes an integration of UAV system for detailed documnetaiton of architectural and archeological heritage in areas of Rocca Farnese in Capodimonte. In the second part, an instrumental survey for the creation of a realistic digital twin is conducted with an aim to provide an immersive VR experience for the archaeological area of Santa Croce in Gerusalemme in Rome.

The title of the paper is too long. It should be a concise and informative representation of the paper's content.

Figure 1 is hard to read. It would be better to have larger fonts and put the taxonomy vertically instead of horizontally to gain more space.

Table 1 provides a summary of survey parameters and output data of each sampling mission. The references to drones/UAVs used in the survey should be included in the text for the readers to explore and co-relate their flight endurance and payload capacity for a similar study.

In the end, a conclusion and future direction of the reserach study must be included.

The paper presents an applicaiton of VR to create a digital twin for virtual tours. However, far more benefits can be obtained such as:

a) Restoration and Reconstruction in future

b) Risk assessment and Conservation Planning

c) Archaeological Excavations and Investigations

d) Data storage and collaboraiton

to name few. These possible benefits must be highlighted in the text with relevant references.

Author Response

REVIEWER 3

This paper describes an integration of UAV system for detailed documnetaiton of architectural and archeological heritage in areas of Rocca Farnese in Capodimonte. In the second part, an instrumental survey for the creation of a realistic digital twin is conducted with an aim to provide an immersive VR experience for the archaeological area of Santa Croce in Gerusalemme in Rome.

The title of the paper is too long. It should be a concise and informative representation of the paper's content.

from

“Integrated survey for a detailed documentation of architectural and archeological heritage. Integration of UAV systems, total stations, terrestrial photogrammetry, and laser scanning”

to

“Integrated survey, from laser scanner to UAV System, for a detailed documentation of architectural and archeological heritage

Figure 1 is hard to read. It would be better to have larger fonts and put the taxonomy vertically instead of horizontally to gain more space.

We changed the figure 1 with larger fonts.

Table 1 provides a summary of survey parameters and output data of each sampling mission. The references to drones/UAVs used in the survey should be included in the text for the readers to explore and co-relate their flight endurance and payload capacity for a similar study.

We added on the text the specific flights done by each drone (Lines 565-567).

In the end, a conclusion and future direction of the reserach study must be included.

We added a critical though about the future direction e research. Including also some of the application of the digital twin now and in the future (lines 779- 864)

The paper presents an applicaiton of VR to create a digital twin for virtual tours. However, far more benefits can be obtained such as:

  1. a) Restoration and Reconstruction in future
  2. b) Risk assessment and Conservation Planning
  3. c) Archaeological Excavations and Investigations
  4. d) Data storage and collaboraiton

to name few. These possible benefits must be highlighted in the text with relevant references.

We decided to cite the suggested cases to enrich the references to cases related to the work presented in the paper, underlining how similar approaches can answer different goals (lines 154-166)

Round 2

Reviewer 2 Report

The manuscript is well revised and it is acceptable in its current form.

The manuscript is well revised and it is acceptable in its current form.